# Near-Optimal No-Regret Learning in General Games

**Constantinos Daskalakis**
MIT CSAIL
costis@csail.mit.edu

**Maxwell Fishelson**
MIT CSAIL
maxfish@mit.edu

**Noah Golowich**
MIT CSAIL
nzg@mit.edu

## Abstract

We show that Optimistic Hedge – a common variant of multiplicative-weights-updates with recency bias – attains $\text{poly}(\log T)$ regret in multi-player general-sum games. In particular, when every player of the game uses Optimistic Hedge to iteratively update her strategy in response to the history of play so far, then after $T$ rounds of interaction, *each player* experiences total regret that is $\text{poly}(\log T)$. Our bound improves, exponentially, the $O(T^{1/2})$ regret attainable by standard no-regret learners in games, the $O(T^{1/4})$ regret attainable by no-regret learners with recency bias [SALS15], and the $O(T^{1/6})$ bound that was recently shown for Optimistic Hedge in the special case of two-player games [CP20]. A corollary of our bound is that Optimistic Hedge converges to coarse correlated equilibrium in general games at a rate of $\tilde{O}\left(\frac{1}{T}\right)$.

## 1 Introduction

Online learning has a long history that is intimately related to the development of game theory, convex optimization, and machine learning. One of its earliest instantiations can be traced to Brown's proposal [Bro49] of fictitious play as a method to solve two-player zero-sum games. Indeed, as shown by [Rob51], when the players of (zero-sum) matrix game use fictitious play to iteratively update their actions in response to each other's history of play, the resulting dynamics converge in the following sense: the product of the empirical distributions of strategies for each player converges to the set of Nash equilibria in the game, though the rate of convergence is now known to be exponentially slow [DP14]. Moreover, such convergence to Nash equilibria fails in non-zero-sum games [Sha64].

The slow convergence of fictitious play to Nash equilibria in zero-sum matrix games and non-convergence in general-sum games can be mitigated by appealing to the pioneering works [Bla54, Han57] and the ensuing literature on no-regret learning [CBL06]. It is known that if both players of a zero-sum matrix game experience regret that is at most $\varepsilon(T)$, the product of the players' empirical distributions of strategies is an $O(\varepsilon(T)/T)$-approximate Nash equilibrium. More generally, if each player of a general-sum, multi-player game experiences regret that is at most $\varepsilon(T)$, the empirical distribution of joint strategies converges to a coarse correlated equilibrium[1] of the game, at a rate of $O(\varepsilon(T)/T)$. Importantly, a multitude of online learning algorithms, such as the celebrated Hedge and Follow-The-Perturbed-Leader algorithms, guarantee adversarial regret $O(\sqrt{T})$ [CBL06]. Thus, when such algorithms are employed by all players in a game, their $O(\sqrt{T})$ regret implies convergence to coarse correlated equilibria (and Nash equilibria of matrix games) at a rate of $O(1/\sqrt{T})$.

While standard no-regret learners guarantee $O(\sqrt{T})$ regret for each player in a game, the players can do better by employing specialized no-regret learning procedures. Indeed, it was established by [DDK11] that there exists a somewhat complex no-regret learner based on Nesterov's excessive gap technique [Nes05], which guarantees $O(\log T)$ regret to each player of a two-player zero-sum game.

---

[1]In general-sum games, it is typical to focus on proving convergence rates for weaker types of equilibrium than Nash, such as coarse correlated equilibria, since finding Nash equilibria is PPAD-complete [DGP06, CDT09].

Table 1: Overview of prior work on fast rates for learning in games. $m$ denotes the number of players, and $n$ denotes the number of actions per player (assumed to be the same for all players). For Optimistic Hedge, the adversarial regret bounds in the right-hand column are obtained via a choice of adaptive step-sizes. The $\tilde{O}(\cdot)$ notation hides factors that are polynomial in $\log T$.

| Algorithm | Setting | Regret in games | Adversarial regret |
|---|---|---|---|
| Hedge (& many other algs.) | multi-player, general-sum | $O(\sqrt{T \log n})$ [CBL06] | $O(\sqrt{T \log n})$ [CBL06] |
| Excessive Gap Technique | 2-player, 0-sum | $O(\log n(\log T + \log^{3/2} n))$ [DDK11] | $O(\sqrt{T \log n})$ [DDK11] |
| DS-OptMD, OptDA | 2-player, 0-sum | $\log^{O(1)}(n)$ [HAM21] | $\sqrt{T \log^{O(1)}(n)}$ [HAM21] |
| Optimistic Hedge | multi-player, general-sum | $O(\log n \cdot \sqrt{m} \cdot T^{1/4})$ [RS13b, SALS15] | $\tilde{O}(\sqrt{T \log n})$ [RS13b, SALS15] |
| Optimistic Hedge | 2-player, general-sum | $O(\log^{5/6} n \cdot T^{1/6})$ [CP20] | $\tilde{O}(\sqrt{T \log n})$ |
| Optimistic Hedge | multi-player, general-sum | $O(\log n \cdot m \cdot \log^4 T)$ **(Theorem 3.1)** | $\tilde{O}(\sqrt{T \log n})$ **(Corollary D.1)** |

This represents an exponential improvement over the regret guaranteed by standard no-regret learners. More generally, [SALS15] established that if players of a multi-player, general-sum game use any algorithm from the family of Optimistic Mirror Descent (MD) or Optimistic Follow-the-Regularized-Leader (FTRL) algorithms (which are analogues of the MD and FTRL algorithms, respectively, with recency bias), each player enjoys regret that is $O(T^{1/4})$. This was recently improved by [CP20] to $O(T^{1/6})$ in the special case of two-player games in which the players use Optimistic Hedge, a particularly simple representative from both the Optimistic MD and Optimistic FTRL families.

The above results for general-sum games represent significant improvements over the $O(\sqrt{T})$ regret attainable by standard no-regret learners, but are not as dramatic as the logarithmic regret that has been shown attainable by no-regret learners, albeit more complex ones, in 2-player zero-sum games (e.g., [DDK11]). Indeed, despite extensive work on no-regret learning, understanding the optimal regret that can be guaranteed by no-regret learning algorithms in general-sum games has remained elusive. This question is especially intruiging in light of experiments suggesting that polylogarithmic regret should be attainable [SALS15, HAM21]. In this paper we settle this question by showing that no-regret learners can guarantee polylogarithmic regret to each player in general-sum multi-player games. Moreover, this regret is attainable by a particularly simple algorithm – Optimistic Hedge:

**Theorem 1.1** (Abbreviated version of Theorem 3.1)**.** *Suppose that $m$ players play a general-sum multi-player game, with a finite set of $n$ strategies per player, over $T$ rounds. Suppose also that each player uses Optimistic Hedge to update her strategy in every round, as a function of the history of play so far. Then each player experiences $O(m \cdot \log n \cdot \log^4 T)$ regret.*

An immediate corollary of Theorem 1.1 is that the empirical distribution of play is a $O\left(\frac{m \log n \log^4 T}{T}\right)$-approximate coarse correlated equilibrium (CCE) of the game. We remark that Theorem 1.1 bounds the total regret experienced by *each player* of the multi-player game, which is the most standard regret objective for no-regret learning in games, and which is essential to achieve convergence to CCE. For the looser objective of the *average* of all players' regrets, [RS13b] established a $O(\log n)$ bound for Optimistic Hedge in two-player zero-sum games, and [SALS15] generalized this bound, to $O(m \log n)$ in $m$-player general-sum games. Note that since some players may experience *negative regret* [HAM21], the average of the players' regrets cannot be used in general to bound the maximum regret experienced by any individual player. Finally, we remark that several results in the literature posit no-regret learning as a model of agents' rational behavior; for instance, [Rou09, ST13, RST17] show that no-regret learners in smooth games enjoy strong Price-of-Anarchy bounds. By showing that *each agent* can obtain very small regret in games by playing Optimistic Hedge, Theorem 1.1

strengthens the plausability of the common assumption made in this literature that each agent will choose to use such a no-regret algorithm.

## 1.1 Related work

Table 1 summarizes the prior works that aim to establish optimal regret bounds for no-regret learners in games. We remark that [CP20] shows that the regret of Hedge is $\Omega(\sqrt{T})$ even in 2-player games where each player has 2 actions, meaning that optimism is necessary to obtain fast rates. The table also includes a recent result of [HAM21] showing that when the players in a 2-player zero-sum game with $n$ actions per player use a variant of Optimistic Hedge with adaptive step size (a special case of their algorithms DS-OptMD and OptDA), each player has $\log^{O(1)} n$ regret. The techniques of [HAM21] differ substantially from ours: the result in [HAM21] is based on showing that the joint strategies $x^{(t)}$ rapidly converge, pointwise, to a Nash equilibrium $x^\star$. Such a result seems very unlikely to extend to our setting of general-sum games, since finding an approximate Nash equilibrium even in 2-player games is PPAD-complete [CDT09]. We also remark that the earlier work [KHSC18] shows that each player's regret is at most $O(\log T \cdot \log n)$ when they use a certain algorithm based on Optimistic MD in 2-player zero-sum games; their technique is heavily tailored to 2-player zero-sum games, relying on the notion of duality in such a setting.

[FLL$^+$16] shows that one can obtain fast rates in games for a broader class of algorithms (e.g., including Hedge) if one adopts a relaxed (approximate) notion of optimality. [WL18] uses optimism to obtain adaptive regret bounds for bandit problems. Many recent papers (e.g., [DP19, GPD20, LGNPw21, HAM21, WLZL21, AIMM21]) have studied the *last-iterate* convergence of algorithms from the Optimistic Mirror Descent family, which includes Optimistic Hedge. Finally, a long line of papers (e.g., [HMcW$^+$03, DFP$^+$10, KLP11, BCM12, PP16, BP18, MPP18, BP19, CP19, VGFL$^+$20]) has studied the dynamics of learning algorithms in games. Essentially all of these papers do not use optimism, and many of them show *non-convergence* (e.g., divergence or recurrence) of the iterates of various learning algorithms such as FTRL and Mirror Descent when used in games.

## 2 Preliminaries

**Notation.** For a positive integer $n$, let $[n] := \{1, 2, \ldots, n\}$. For a finite set $\mathcal{S}$, let $\Delta(\mathcal{S})$ denote the space of distributions on $\mathcal{S}$. For $\mathcal{S} = [n]$, we will write $\Delta^n := \Delta(\mathcal{S})$ and interpret elements of $\Delta^n$ as vectors in $\mathbb{R}^n$. For a vector $v \in \mathbb{R}^n$ and $j \in [n]$, we denote the $j$th coordinate of $v$ as $v(j)$. For vectors $v, w \in \mathbb{R}^n$, write $\langle v, w \rangle = \sum_{j=1}^n v(j)w(j)$. The base-2 logarithm of $x > 0$ is denoted $\log x$.

**No-regret learning in games.** We consider a game $G$ with $m \in \mathbb{N}$ players, where player $i \in [m]$ has *action space* $\mathcal{A}_i$ with $n_i := |\mathcal{A}_i|$ actions. We may assume that $\mathcal{A}_i = [n_i]$ for each player $i$. The *joint action space* is $\mathcal{A} := \mathcal{A}_1 \times \cdots \times \mathcal{A}_m$. The specification of the game $G$ is completed by a collection of *loss functions* $\mathcal{L}_1, \ldots, \mathcal{L}_m : \mathcal{A} \to [0, 1]$. For an action profile $a = (a_1, \ldots, a_m) \in \mathcal{A}$ and $i \in [m]$, $\mathcal{L}_i(a)$ is the loss player $i$ experiences when each player $i' \in [m]$ plays $a_{i'}$. A *mixed strategy* $x_i \in \Delta(\mathcal{A}_i)$ for player $i$ is a distribution over $\mathcal{A}_i$, with the probability of playing action $j \in \mathcal{A}_i$ given by $x_i(j)$. Given a mixed strategy profile $x = (x_1, \ldots, x_m)$ (or an action profile $a = (a_1, \ldots, a_m)$) and a player $i \in [m]$ we let $x_{-i}$ (or $a_{-i}$, respectively) denote the profile after removing the $i$th mixed strategy $x_i$ (or the $i$th action $a_i$, respectively).

The $m$ players play the game $G$ for a total of $T$ rounds. At the beginning of each round $t \in [T]$, each player $i$ chooses a mixed strategy $x_i^{(t)} \in \Delta(\mathcal{A}_i)$. The *loss vector* of player $i$, denoted $\ell_i^{(t)} \in [0, 1]^{n_i}$, is defined as $\ell_i^{(t)}(j) = \mathbb{E}_{a_{-i} \sim x_{-i}^{(t)}}[\mathcal{L}_i(j, a_{-i})]$. As a matter of convention, set $\ell_i^{(0)} = \mathbf{0}$ to be the all-zeros vector. We consider the *full-information setting* in this paper, meaning that player $i$ observes its full loss vector $\ell_i^{(t)}$ for each round $t$. Finally, player $i$ experiences a loss of $\langle \ell_i^{(t)}, x_i^{(t)} \rangle$. The goal of each player $i$ is to minimize its *regret*, defined as: $\text{Reg}_{i,T} := \sum_{t \in [T]} \langle x_i^{(t)}, \ell_i^{(t)} \rangle - \min_{j \in [n_i]} \sum_{t \in [T]} \ell_i^{(t)}(j)$.

**Optimistic hedge.** The Optimistic Hedge algorithm chooses mixed strategies for player $i \in [m]$ as follows: at time $t = 1$, it sets $x_i^{(1)} = (1/n_i, \ldots, 1/n_i)$ to be the uniform distribution on $\mathcal{A}_i$. Then for all $t < T$, player $i$'s strategy at iteration $t + 1$ is defined as follows, for $j \in [n_i]$:

$$x_i^{(t+1)}(j) := \frac{x_i^{(t)}(j) \cdot \exp(-\eta \cdot (2\ell_i^{(t)}(j) - \ell_i^{(t-1)}(j)))}{\sum_{k \in [n_i]} x_i^{(t)}(k) \cdot \exp(-\eta \cdot (2\ell_i^{(t)}(k) - \ell_i^{(t-1)}(k)))}. \tag{1}$$

Optimistic Hedge is a modification of Hedge, which performs the updates $x_i^{(t+1)}(j) :=$ $\frac{x_i^{(t)}(j) \cdot \exp(-\eta \cdot \ell_i^{(t)}(j))}{\sum_{k \in [n_i]} x_i^{(t)}(k) \cdot \exp(-\eta \cdot \ell_i^{(t)}(k))}$. The update (1) modifies the Hedge update by replacing the loss vector $\ell_i^{(t)}$ with a predictor of the *following iteration's* loss vector, $\ell_i^{(t)} + (\ell_i^{(t)} - \ell_i^{(t-1)})$. Hedge corresponds to FTRL with a negative entropy regularizer (see, e.g., [Bub15]), whereas Optimistic Hedge corresponds to *Optimistic* FTRL with a negative entropy regularizer [RS13b, RS13a].

**Distributions & divergences.** For distributions $P, Q$ on a finite domain $[n]$, the *KL divergence* between $P, Q$ is $\mathrm{KL}(P; Q) = \sum_{j=1}^{n} P(j) \cdot \log\left(\frac{P(j)}{Q(j)}\right)$. The *chi-squared divergence* between $P, Q$ is $\chi^2(P; Q) = \sum_{j=1}^{n} Q(j) \cdot \left(\frac{P(j)}{Q(j)}\right)^2 - 1 = \sum_{j=1}^{n} \frac{(P(j) - Q(j))^2}{Q(j)}$. For a distribution $P$ on $[n]$ and a vector $v \in \mathbb{R}^n$, we write $\mathrm{Var}_P(v) := \sum_{j=1}^{n} P(j) \cdot \left(v(j) - \sum_{k=1}^{n} P(k)v(k)\right)^2$. Also define $\|v\|_P := \sqrt{\sum_{j=1}^{n} P(j) \cdot v(j)^2}$. If further $P$ has full support, then define $\|v\|_P^\star = \sqrt{\sum_{j=1}^{n} \frac{v(j)^2}{P(j)}}$. The above notations will often be used when $P$ is the mixed strategy profile $x_i$ for some player $i$ and $v$ is a loss vector $\ell_i$; in such a case the norms $\|v\|_P$ and $\|v\|_P^\star$ are often called *local norms*.

## 3 Results

Below we state our main theorem, which shows that when all players in a game play according to Optimistic Hedge with appropriate step size, they all experience polylogarithmic individual regrets.

**Theorem 3.1** (Formal version of Theorem 1.1). *There are constants $C, C' > 1$ so that the following holds. Suppose a time horizon $T \in \mathbb{N}$ and a game $G$ with $m$ players and $n_i$ actions for each player $i \in [m]$ is given. Suppose all players play according to Optimistic Hedge with any positive step size $\eta \leq \frac{1}{C \cdot m \log^4 T}$. Then for any $i \in [m]$, the regret of player $i$ satisfies*

$$\mathrm{Reg}_{i,T} \leq \frac{\log n_i}{\eta} + C' \cdot \log T. \tag{2}$$

*In particular, if the players' step size is chosen as $\eta = \frac{1}{C \cdot m \log^4 T}$, then the regret of player $i$ satisfies*

$$\mathrm{Reg}_{i,T} \leq O\left(m \cdot \log n_i \cdot \log^4 T\right). \tag{3}$$

A common goal in the literature on learning in games is to obtain an algorithm that achieves fast rates whan played by all players, and so that each player $i$ still obtains the optimal rate of $O(\sqrt{T})$ in the adversarial setting (i.e., when $i$ receives an arbitrary sequence of losses $\ell_i^{(1)}, \ldots, \ell_i^{(T)}$). We show in Corollary D.1 (in the appendix) that by running Optimistic Hedge with an adaptive step size, this is possible. Table 1 compares our regret bounds discussed in this section to those of prior work.

## 4 Proof overview

In this section we overview the proof of Theorem 3.1; the full proof may be found in the appendix.

### 4.1 New adversarial regret bound

The first step in the proof of Theorem 3.1 is to prove a new regret bound (Lemma 4.1 below) for Optimistic Hedge that holds for an adversarial sequence of losses. We will show in later sections that when *all* players play according to Optimistic Hedge, the right-hand side of the regret bound (4) is bounded by a quantity that grows only poly-logarithmically in $T$.

**Lemma 4.1.** *There is a constant $C > 0$ so that the following holds. Suppose any player $i \in [m]$ follows the Optimistic Hedge updates (1) with step size $\eta < 1/C$, for an arbitrary sequence of losses $\ell_i^{(1)}, \ldots, \ell_i^{(T)} \in [0, 1]^{n_i}$. Then*

$$\mathrm{Reg}_{i,T} \leq \frac{\log n_i}{\eta} + \sum_{t=1}^{T} \left(\frac{\eta}{2} + C\eta^2\right) \mathrm{Var}_{x_i^{(t)}}\left(\ell_i^{(t)} - \ell_i^{(t-1)}\right) - \sum_{t=1}^{T} \frac{(1 - C\eta)\eta}{2} \cdot \mathrm{Var}_{x_i^{(t)}}\left(\ell_i^{(t-1)}\right). \tag{4}$$

The detailed proof of Lemma 4.1 can be found in Section A, but we sketch the main steps here. The starting point is a refinement of [RS13a, Lemma 3] (stated as Lemma A.5), which gives an upper

bound for $\mathrm{Reg}_{i,T}$ in terms of local norms corresponding to each of the iterates $x_i^{(t)}$ of Optimistic Hedge. The bound involves the difference between the Optimistic Hedge iterates $x_i^{(t)}$ and iterates $\tilde{x}_i^{(t)}$ defined by $\tilde{x}_i^{(t)} = \frac{x_i^{(t)}(j) \cdot \exp(-\eta \cdot (\ell_i^{(t)}(j) - \ell_i^{(t-1)}(j)))}{\sum_{k \in [n_i]} x_i^{(t)}(k) \cdot \exp(-\eta \cdot (\ell_i^{(t)}(k) - \ell_i^{(t-1)}(k)))}$:

$$\mathrm{Reg}_{i,T} \le \frac{\log n_i}{\eta} + \sum_{t=1}^{T} \left\| x_i^{(t)} - \tilde{x}_i^{(t)} \right\|_{x_i^{(t)}}^{\star} \sqrt{\mathrm{Var}_{x_i^{(t)}} \left( \ell_i^{(t)} - \ell_i^{(t-1)} \right)} - \frac{1}{\eta} \sum_{t=1}^{T} \mathrm{KL}(\tilde{x}_i^{(t)}; x_i^{(t)}) - \frac{1}{\eta} \sum_{t=1}^{T} \mathrm{KL}(x_i^{(t)}; \tilde{x}_i^{(t-1)}).$$

(5)

We next show (in Lemma A.2) that $\mathrm{KL}(\tilde{x}_i^{(t)}; x_i^{(t)})$ and $\mathrm{KL}(x_i^{(t)}; \tilde{x}_i^{(t-1)})$ may be lower bounded by $(1/2 - O(\eta)) \cdot \chi^2(\tilde{x}_i^{(t)}; x_i^{(t)})$ and $(1/2 - O(\eta)) \cdot \chi^2(x_i^{(t)}; \tilde{x}_i^{(t-1)})$, respectively. Note it is a standard fact that the KL divergence between two distributions is upper bounded by the chi-squared distribution between them; by contrast, Lemma A.2 can exploit that $x_i^{(t)}$, $\tilde{x}_i^{(t)}$ and $\tilde{x}_i^{(t-1)}$ are close to each other to show a reverse inequality. Finally, exploiting the exponential weights-style functional relationship between $x_i^{(t)}$ and $\tilde{x}_i^{(t-1)}$, we show (in Lemma A.3) that the $\chi^2$-divergence $\chi^2(x_i^{(t)}; \tilde{x}_i^{(t-1)})$ may be lower bounded by $(1 - O(\eta)) \cdot \eta^2 \cdot \mathrm{Var}_{x_i^{(t)}} \left( \ell_i^{(t-1)} \right)$, leading to the term $\frac{(1-C\eta)\eta}{2} \mathrm{Var}_{x_i^{(t)}} \left( \ell_i^{(t-1)} \right)$ being subtracted in (4). The $\chi^2$-divergence $\chi^2(\tilde{x}_i^{(t)}; x_i^{(t)})$, as well as the term $\left\| x_i^{(t)} - \tilde{x}_i^{(t)} \right\|_{x_i^{(t)}}^{\star}$ in (5) are bounded in a similar manner to obtain (4).

## 4.2 Finite differences

Given Lemma 4.1, in order to establish Theorem 3.1, it suffices to show Lemma 4.2 below. Indeed, (6) below implies that the right-hand side of (4) is bounded above by $\frac{\log n_i}{\eta} + \eta \cdot O(\log^5 T)$, which is bounded above by $O(m \log n_i \log^4 T)$ for the choice $\eta = \Theta\left( \frac{1}{m \cdot \log^4 T} \right)$ of Theorem 3.1.[2]

**Lemma 4.2** (Abbreviated; detailed version in Section C.3)**.** *Suppose all players play according to Optimistic Hedge with step size $\eta$ satifying $1/T \le \eta \le \frac{1}{Cm \cdot \log^4 T}$ for a sufficiently large constant $C$. Then for any $i \in [m]$, the losses $\ell_i^{(1)}, \ldots, \ell_i^{(T)} \in \mathbb{R}^{n_i}$ for player $i$ satisfy:*

$$\sum_{t=1}^{T} \mathrm{Var}_{x_i^{(t)}} \left( \ell_i^{(t)} - \ell_i^{(t-1)} \right) \le \frac{1}{2} \cdot \sum_{t=1}^{T} \mathrm{Var}_{x_i^{(t)}} \left( \ell_i^{(t-1)} \right) + O\left( \log^5 T \right).$$

(6)

The definition below allows us to streamline our notation when proving Lemma 4.2.

**Definition 4.1** (Finite differences)**.** Suppose $L = (L^{(1)}, \ldots, L^{(T)})$ is a sequence of vectors $L^{(t)} \in \mathbb{R}^n$. For integers $h \ge 0$, the *order-h finite difference sequence* for the sequence $L$, denoted by $\mathrm{D}_h L$, is the sequence $\mathrm{D}_h L := ((\mathrm{D}_h L)^{(1)}, \ldots, (\mathrm{D}_h L)^{(T-h)})$ defined recursively as: $(\mathrm{D}_0 L)^{(t)} := L^{(t)}$ for all $1 \le t \le T$, and

$$(\mathrm{D}_h L)^{(t)} := (\mathrm{D}_{h-1} L)^{(t+1)} - (\mathrm{D}_{h-1} L)^{(t)}$$

(7)

for all $h \ge 1$, $1 \le t \le T - h$.[3]

**Remark 4.3.** *Notice that another way of writing (7) is: $\mathrm{D}_h L = \mathrm{D}_1 \mathrm{D}_{h-1} L$. We also remark for later use that $(\mathrm{D}_h L)^{(t)} = \sum_{s=0}^{h} \binom{h}{s} (-1)^{h-s} L^{(t+s)}$.*

Let $H = \log T$, where $T$ denotes the fixed time horizon from Theorem 3.1 (and thus Lemma 4.2). In the proof of Lemma 4.2, we will bound the finite differences of order $h \le H$ for certain sequences. The bound (6) of Lemma 4.2 may be rephased as upper bounding $\sum_{t=1}^{T} \mathrm{Var}_{x_i^{(t)}} \left( (\mathrm{D}_1 \ell_i)^{(t-1)} \right)$, by $\frac{1}{2} \sum_{t=1}^{T} \mathrm{Var}_{x_i} \left( \ell_i^{(t-1)} \right)$; to prove this, we proceed in two steps:

---

[2]Notice that the factor $\frac{1}{2}$ in (6) is not important for this argument – any constant less than 1 would suffice.

[3]We remark that while Definition 4.1 is stated for a 1-indexed sequence $L^{(1)}, L^{(2)}, \ldots$, we will also occasionally consider 0-indexed sequences $L^{(0)}, L^{(1)}, \ldots$, in which case the same recursive definition (7) holds for the finite differences $(\mathrm{D}_h L)^{(t)}$, $t \ge 0$.

1. (*Upwards induction step*) First, in Lemma 4.4 below, we find an upper bound on $\left\|(D_h\,\ell_i)^{(t)}\right\|_\infty$ for all $t \in [T]$, $h \geq 0$, which decays exponentially in $h$ for $h \leq H$. This is done via *upwards induction* on $h$, i.e., first proving the base case $h = 0$ using boundedness of the losses $\ell_i^{(t)}$ and then $h = 1, 2, \ldots$ inductively. The main technical tool we develop for the inductive step is a weak form of the chain rule for finite differences, Lemma 4.5. The inductive step uses the fact that all players are following Optimistic Hedge to relate the $h$th order finite differences of player $i$'s loss sequence $\ell_i^{(t)}$ to the $h$th order finite differences of the strategy sequences $x_{i'}^{(t)}$ for players $i' \neq i$; then we use the exponential-weights style updates of Optimistic Hedge and Lemma 4.5 to relate the $h$th order finite differences of the strategies $x_{i'}^{(t)}$ to the $(h-1)$th order finite differences of the losses $\ell_{i'}^{(t)}$.

2. (*Downwards induction step*) We next show that for all $0 \leq h \leq H$, $\sum_{t=1}^T \mathrm{Var}_{x_i^{(t)}}\left((D_{h+1}\,\ell_i)^{(t-1)}\right)$ is bounded above by $c_h \cdot \sum_{t=1}^T \mathrm{Var}_{x_i^{(t)}}\left((D_h\,\ell_i)^{(t-1)}\right) + \mu_h$, for some $c_h < 1/2$ and $\mu_h < O(\log^5 T)$. This shown via *downwards induction* on $h$, namely first establishing the base case $h = H$ by using the result of item 1 for $h = H$ and then treating the cases $h = H-1, H-2, \ldots, 0$. The inductive step makes use of the discrete Fourier transform (DFT) to relate the finite differences of different orders (see Lemmas 4.7 and 4.8). In particular, Parseval's equality together with a standard relationship between the DFT of the finite differences of a sequence to the DFT of that sequence allow us to first prove the inductive step in the frequency domain and then transport it back to the original (time) domain.

In the following subsections we explain in further detail how the two steps above are completed.

## 4.3 Upwards induction proof overview

Addressing item 1 in the previous subsection, the lemma below gives a bound on the supremum norm of the $h$-th order finite differences of each player's loss vector, when all players play according to Optimistic Hedge and experience losses according to their loss functions $\mathcal{L}_1, \ldots, \mathcal{L}_m : \mathcal{A} \to [0, 1]$.

**Lemma 4.4** (Abbreviated). *Fix a step size* $\eta > 0$ *satisfying* $\eta \leq o\left(\frac{1}{m \log T}\right)$. *If all players follow Optimistic Hedge updates with step size* $\eta$, *then for any player* $i \in [m]$, *integer* $h$ *satisfying* $0 \leq h \leq H$, *and time step* $t \in [T - h]$, *it holds that* $\|(D_h\,\ell_i)^{(t)}\|_\infty \leq O(m\eta)^h \cdot h^{O(h)}$.

A detailed version of Lemma 4.4, together with its full proof, may be found in Section B.4. We next give a proof overview of Lemma 4.4 for the case of 2 players, i.e., $m = 2$; we show in Section B.4 how to generalize this computation to general $m$. Below we introduce the main technical tool in the proof, a "boundedness chain rule," and then outline how it is used to prove Lemma 4.4.

**Main technical tool for Lemma 4.4: boundedness chain rule.** We say that a function $\phi : \mathbb{R}^n \to \mathbb{R}$ is a *softmax-type* function if there are real numbers $\xi_1, \ldots, \xi_n$ and some $j \in [n]$ so that for all $(z_1, \ldots, z_n) \in \mathbb{R}^n$, $\phi((z_1, \ldots, z_n)) = \frac{\exp(z_j)}{\sum_{k=1}^n \xi_k \cdot \exp(z_k)}$. Lemma 4.5 below may be interpreted as a "boundedness chain rule" for finite differences. To explain the context for this lemma, recall that given an infinitely differentiable vector-valued function $L : \mathbb{R} \to \mathbb{R}^n$ and an infinitely differentiable function $\phi : \mathbb{R}^n \to \mathbb{R}$, the higher order derivatives of the function $\phi(L(t))$ may be computed in terms of those of $L$ and $\phi$ using the chain rule. Lemma 4.5 considers an analogous setting where the input variable $t$ to $L$ is discrete-valued, taking values in $[T]$ (and so we identify the function $L$ with the sequence $L^{(1)}, \ldots, L^{(T)}$). In this case, the *higher order finite differences* of the sequence $L^{(1)}, \ldots, L^{(T)}$ (Definition 4.1) take the place of the higher order derivatives of $L$ with respect to $t$. Though there is no generic chain rule for finite differences, Lemma 4.5 states that, at least when $\phi$ is a softmax-type function, we may *bound* the higher order finite differences of the sequence $\phi(L^{(1)}), \ldots, \phi(L^{(T)})$. In the lemma's statement we let $\phi \circ L$ denote the sequence $\phi(L^{(1)}), \ldots, \phi(L^{(T)})$.

**Lemma 4.5** ("Boundedness chain rule" for finite differences; abbreviated). *Suppose that* $h, n \in \mathbb{N}$, $\phi : \mathbb{R}^n \to \mathbb{R}$ *is a softmax-type function, and* $L = (L^{(1)}, \ldots, L^{(T)})$ *is a sequence of vectors in* $\mathbb{R}^n$ *satisfying* $\|L^{(t)}\|_\infty \leq 1$ *for* $t \in [T]$. *Suppose for some* $\alpha \in (0, 1)$, *for each* $0 \leq h' \leq h$ *and* $t \in [T - h']$, *it holds that* $\|D_{h'}\,L^{(t)}\|_\infty \leq O(\alpha^{h'}) \cdot (h')^{O(h')}$. *Then for all* $t \in [T - h]$,
$$|(D_h\,(\phi \circ L))^{(t)}| \leq O(\alpha^h) \cdot h^{O(h)}.$$

A detailed version of Lemma 4.5 may be found in Section B.3. While Lemma 4.5 requires $\phi$ to be a softmax-type function for simplicity (and this is the only type of function $\phi$ we will need to consider

for the case $m = 2$) we remark that the detailed version of Lemma 4.5 allows $\phi$ to be from a more general family of analytic functions whose higher order derivatives are appropriately bounded. The proof of Lemma 4.4 for all $m \geq 2$ requires that more general form of Lemma 4.5.

The proof of Lemma 4.5 proceeds by considering the Taylor expansion $P_\phi(\cdot)$ of the function $\phi$ at the origin, which we write as follows: for $z = (z_1, \ldots, z_n) \in \mathbb{R}^n$, $P_\phi(z) := \sum_{k \geq 0, \gamma \in \mathbb{Z}_{\geq 0}^n: |\gamma| = k} a_\gamma z^\gamma$, where $a_\gamma \in \mathbb{R}$, $|\gamma|$ denotes the quantity $\gamma_1 + \cdots + \gamma_n$ and $z^\gamma$ denotes $z_1^{\gamma_1} \cdots z_n^{\gamma_n}$. The fact that $\phi$ is a softmax-type function ensures that the radius of convergence of its Taylor series is at least 1, i.e., $\phi(z) = P_\phi(z)$ for any $z$ satisfying $\|z\|_\infty \leq 1$. By the assumption that $\|L^{(t)}\|_\infty \leq 1$ for each $t$, we may therefore decompose $(D_h (\phi \circ L))^{(t)}$ as:

$$(D_h (\phi \circ L))^{(t)} = \sum_{k \geq 0, \gamma \in \mathbb{Z}_{\geq 0}^n: |\gamma| = k} a_\gamma \cdot (D_h L^\gamma)^{(t)}, \tag{8}$$

where $L^\gamma$ denotes the sequence of scalars $(L^\gamma)^{(t)} := (L^{(t)})^\gamma$ for all $t$. The fact that $\phi$ is a softmax-type function allows us to establish strong bounds on $|a_\gamma|$ for each $\gamma$ in Lemma B.5. The proof of Lemma B.5 bounds the $|a_\gamma|$ by exploiting the simple form of the derivative of a softmax-type function to decompose each $a_\gamma$ into a sum of $|\gamma|!$ terms. Then we establish a bijection between the terms of this decomposition and graph structures we refer to as *factorial trees*; that bijection together with the use of an appropriate generating function allow us to complete the proof of Lemma B.5.

Thus, to prove Lemma 4.5, it suffices to bound $\left|(D_h L^\gamma)^{(t)}\right|$ for all $\gamma$. We do so by using Lemma 4.6.

**Lemma 4.6** (Abbreviated; detailed vesion in Section B.2). *Fix any $h \geq 0$, a multi-index $\gamma \in \mathbb{Z}_{\geq 0}^n$ and set $k = |\gamma|$. For each of the $k^h$ functions $\pi : [h] \to [k]$, and for each $r \in [k]$, there are integers $h'_{\pi,r} \in \{0, 1, \ldots, h\}$, $t'_{\pi,r} \geq 0$, and $j'_{\pi,r} \in [n]$, so that the following holds. For any sequence $L^{(1)}, \ldots, L^{(T)} \in \mathbb{R}^n$ of vectors, it holds that, for each $t \in [T - h]$,*

$$(D_h L^\gamma)^{(t)} = \sum_{\pi:[h] \to [k]} \prod_{r=1}^k \left(D_{h'_{\pi,r}}(L(j'_{\pi,r}))\right)^{(t+t'_{\pi,r})}. \tag{9}$$

Lemma 4.6 expresses the $h$th order finite differences of the sequence $L^\gamma$ as a sum of $k^h$ terms, each of which is a product of $k$ finite order differences of a sequence $L^{(t)}(j'_{\pi,r})$ (i.e., the $j'_{\pi,r}$th coordinate of the vectors $L^{(t)}$). Crucially, when using Lemma 4.6 to prove Lemma 4.5, the assumption of Lemma 4.5 gives that for each $j' \in [n]$, each $h' \in [h]$, and each $t' \in [T - h']$, we have the bound $\left|(D_{h'} L(j'))^{(t')}\right| \leq O(\alpha^{h'}) \cdot (h')^{O(h')}$. These assumed bounds may be used to bound the right-hand side of (9), which together with Lemma 4.6 and (8) lets us complete the proof of Lemma 4.5.

**Proving Lemma 4.4 using the boundedness chain rule.** Next we discuss how Lemma 4.5 is used to prove Lemma 4.4, namely to bound $\|(D_h \ell_i)^{(t)}\|_\infty$ for each $t \in [T - h]$, $i \in [m]$, and $0 \leq h \leq H$. Lemma 4.4 is proved using induction, with the base case $h = 0$ being a straightforward consequence of the fact that $\|(D_0 \ell_i)^{(t)}\|_\infty = \|\ell_i^{(t)}\|_\infty \leq 1$ for all $i \in [m], t \in [T]$. For the rest of this section we focus on the inductive case, i.e., we pick some $h \in [H]$ and assume Lemma 4.4 holds for all $h' < h$.

The first step is to reduce the claim of Lemma 4.4 to the claim that the upper bound $\|(D_h x_i)^{(t)}\|_1 \leq O(m\eta)^h \cdot h^{O(h)}$ holds for each $t \in [T - h], i \in [m]$. Recalling that we are only sketching here the case $m = 2$ for simplicity, this reduction proceeds as follows: for $i \in \{1, 2\}$, define the matrix $A_i \in \mathbb{R}^{n_1 \times n_2}$ by $(A_i)_{a_1 a_2} = \mathcal{L}_i(a_1, a_2)$, for $a_1 \in [n_1], a_2 \in [n_2]$. We have assumed that all players are using Optimistic Hedge and thus $\ell_i^{(t)} = \mathbb{E}_{a_{i'} \sim x_{i'}^{(t)}, \forall i' \neq i}[\mathcal{L}_i(a_1, \ldots, a_n)]$; for our case here ($m = 2$), this may be rewritten as $\ell_1^{(t)} = A_1 x_2^{(t)}, \ell_2^{(t)} = A_2^\top x_1^{(t)}$. Thus

$$\|(D_h \ell_1)^{(t)}\|_\infty = \left\|A_1 \cdot \sum_{s=0}^h \binom{h}{s}(-1)^{h-s} x_2^{(t+s)}\right\|_\infty \leq \left\|\sum_{s=0}^h \binom{h}{s}(-1)^{h-s} x_2^{(t+s)}\right\|_1 = \|(D_h x_2)^{(t)}\|_1,$$

where the first equality is from Remark 4.3 and the inequality follows since all entries of $A_1$ have absolute value $\leq 1$. A similar computation allows us to show $\|(D_h \ell_2)^{(t)}\|_\infty \leq \|(D_h x_1)^{(t)}\|_1$.

To complete the inductive step it remains to upper bound the quantities $\| (D_h x_i)^{(t)} \|_1$ for $i \in [m], t \in [T-h]$. To do so, we note that the definition of the Optimistic Hedge updates (1) implies that for any $i \in [m], t \in [T], j \in [n_i]$, and $t' \geq 1$, we have

$$x_i^{(t+t')}(j) = \frac{x_i^{(t)}(j) \cdot \exp\left( \eta \cdot \left( \ell_i^{(t-1)}(j) - \sum_{s=0}^{t'-1} \ell_i^{(t+s)}(j) - \ell_i^{(t+t'-1)}(j) \right) \right)}{\sum_{k=1}^{n_i} x_i^{(t)}(k) \cdot \exp\left( \eta \cdot \left( \ell_i^{(t-1)}(k) - \sum_{s=0}^{t'-1} \ell_i^{(t+s)}(k) - \ell_i^{(t+t'-1)}(k) \right) \right)}. \quad (10)$$

For $t \in [T], t' \geq 0$, set $\bar{\ell}_{i,t}^{(t')} := \eta \cdot \left( \ell_i^{(t-1)} - \sum_{s=0}^{t'-1} \ell_i^{(t+s)} - \ell_i^{(t+t'-1)} \right)$. Also, for each $i \in [m], j \in [n_i], t \in [T]$, and any vector $z = (z(1), \ldots, z(n_i)) \in \mathbb{R}^{n_i}$ define $\phi_{t,i,j}(z) := \frac{x_i^{(t)}(j) \cdot \exp(z(j))}{\sum_{k=1}^{n_i} x_i^{(t)}(k) \cdot \exp(z(k))}$. Thus (10) gives that for $t' \geq 1$, $x_i^{(t+t')}(j) = \phi_{t,i,j}(\bar{\ell}_{i,t}^{(t')})$. Viewing $t$ as a fixed parameter and letting $t'$ vary, it follows that for $h \geq 0$ and $t' \geq 1$, $\left( D_h x_i^{(t+\cdot)}(j) \right)^{(t')} = \left( D_h (\phi_{t,i,j} \circ \bar{\ell}_{i,t}) \right)^{(t')}$.

Recalling that our goal is to bound $|(D_h x_i(j))^{(t+1)}|$ for each $t$, we can do so by using Lemma 4.5 with $\phi = \phi_{t,i,j}$ and $\alpha = O(m\eta)$, if we can show that its precondition is met, i.e. that $\| (D_{h'} \bar{\ell}_{i,t})^{(t')} \|_\infty \leq \frac{1}{B_1} \cdot \alpha^{h'} \cdot (h')^{B_0 h'}$ for all $h' \leq h$, the appropriate $\alpha$ and appropriate constants $B_0, B_1$. Helpfully, the definition of $\bar{\ell}_{i,t}^{(t')}$ as a partial sum allows us to relate the $h'$-th order finite differences of the sequence $\bar{\ell}_{i,t}^{(t')}$ to the $(h'-1)$-th order finite differences of the sequence $\ell_i^{(t)}$ as follows:

$$\left( D_{h'} \bar{\ell}_{i,t} \right)^{(t')} = \eta \cdot (D_{h'-1} \ell_i)^{(t+t'-1)} - 2\eta \cdot (D_{h'-1} \ell_i)^{(t+t')}. \quad (11)$$

Since $h' - 1 < h$ for $h' \leq h$, the inductive assumption of Lemma 4.4 gives a bound on the $\ell_\infty$-norm of the terms on the right-hand side of (11), which are sufficient for us to apply Lemma 4.5. Note that the inductive assumption gives an upper bound on $\| (D_{h'-1} \ell_i)^{(t)} \|_\infty$ that only scales with $\alpha^{h'-1}$, whereas Lemma 4.5 requires scaling of $\alpha^{h'}$. This discrepancy is corrected by the factor of $\eta$ on the right-hand side of (11), which gives the desired scaling $\alpha^{h'}$ (since $\eta < \alpha$ for the choice $\alpha = O(m\eta)$).

### 4.4 Downwards induction proof overview

In this section we discuss in further detail item 2 in Section 4.2; in particular, we will show that there is a parameter $\mu = \tilde{\Theta}(\eta m)$ so that for all integers $h$ satisfying $H - 1 \geq h \geq 0$,

$$\sum_{t=1}^{T-h-1} \text{Var}_{x_i^{(t)}} \left( (D_{h+1} \ell_i)^{(t)} \right) \leq O(1/H) \cdot \sum_{t=1}^{T-h} \text{Var}_{x_i^{(t)}} \left( (D_h \ell_i)^{(t)} \right) + \tilde{O}\left( \mu^{2h} \right), \quad (12)$$

where $\tilde{O}$ hides factors polynomial in $\log T$. The validity of (12) for $h = 0$ implies Lemma 4.2. On the other hand, as long we choose the value $\mu$ in (12) to satisfy $\mu \geq m\eta H^{\Omega(1)}$, then Lemma 4.4 implies that $\sum_{t=1}^{T-H} \text{Var}_{x_i^{(t)}} \left( (D_H \ell_i)^{(t)} \right) \leq O(\mu^{2H})$. This gives that (12) holds for $h = H - 1$. To show that (12) holds for all $H - 1 > h \geq 0$, we use downwards induction; fix any $h$, and assume that (12) has been shown for all $h'$ satisfying $h < h' \leq H - 1$. Our main tool in the inductive step is to apply Lemma 4.7 below. To state it, for $\zeta > 0$, $n \in \mathbb{N}$, we say that a sequence of distributions $P^{(1)}, \ldots, P^{(T)} \in \Delta^n$ is $\zeta$-consecutively close if for each $1 \leq t < T$, it holds that $\max\left\{ \left\| \frac{P^{(t)}}{P^{(t+1)}} \right\|_\infty, \left\| \frac{P^{(t+1)}}{P^{(t)}} \right\|_\infty \right\} \leq 1 + \zeta$.[4] Lemma 4.7 shows that given a sequence of vectors for which the variances of its second-order finite differences are bounded by the variances of its first-order finite differences, a similar relationship holds between its first- and zeroth-order finite differences.

**Lemma 4.7.** *There is a sufficiently large constant $C_0 > 1$ so that the following holds. For any $M, \zeta, \alpha > 0$ and $n \in \mathbb{N}$, suppose that $P^{(1)}, \ldots, P^{(T)} \in \Delta^n$ and $Z^{(1)}, \ldots, Z^{(T)} \in [-M, M]^n$ satisfy the following conditions:*

1. *The sequence $P^{(1)}, \ldots, P^{(T)}$ is $\zeta$-consecutively close for some $\zeta \in [1/(2T), \alpha^4/C_0]$.*

2. *It holds that $\sum_{t=1}^{T-2} \text{Var}_{P^{(t)}} \left( (D_2 Z)^{(t)} \right) \leq \alpha \cdot \sum_{t=1}^{T-1} \text{Var}_{P^{(t)}} \left( (D_1 Z)^{(t)} \right) + \mu$.*

---

[4]Here, for distributions $P, Q \in \Delta^n$, $\frac{P}{Q} \in \mathbb{R}^n$ denotes the vector whose $j$th entry is $P(j)/Q(j)$.

*Then* $\sum_{t=1}^{T-1} \mathrm{Var}_{P^{(t)}}\left((\mathrm{D}_1\, Z)^{(t)}\right) \leq \alpha \cdot (1+\alpha) \sum_{t=1}^{T} \mathrm{Var}_{P^{(t)}}\left(Z^{(t)}\right) + \frac{\mu}{\alpha} + \frac{C_0 M^2}{\alpha^3}$.

Given Lemma 4.7, the inductive step for establishing (12) is straightforward: we apply Lemma 4.7 with $P^{(t)} = x_i^{(t)}$ and $Z^{(t)} = (\mathrm{D}_h\,\ell_i)^{(t)}$ for all $t$. The fact that $x_i^{(t)}$ are updated with Optimistic Hedge may be used to establish that precondition 1 of Lemma 4.7 holds. Since $(\mathrm{D}_1\, Z)^{(t)} = (\mathrm{D}_{h+1}\,\ell_i)^{(t)}$ and $(\mathrm{D}_2\, Z)^{(t)} = (\mathrm{D}_{h+2}\,\ell_i)^{(t)}$, that the inductive hypothesis (12) holds for $h+1$ implies that precondition 2 of Lemma 4.7 holds for appropriate $\alpha, \mu > 0$. Thus Lemma 4.7 implies that (12) holds for the value $h$, which completes the inductive step.

**On the proof of Lemma 4.7.** Finally we discuss the proof of Lemma 4.7. One technical challenge is the fact that the vectors $P^{(t)}$ are not constant functions of $t$, but rather change slowly (as constrained by being $\zeta$-consecutively close). The main tool for dealing with this difficulty is Lemma C.1, which shows that for a $\zeta$-consecutively close sequence $P^{(t)}$, for any vector $Z^{(t)}$, $\frac{\mathrm{Var}_{P^{(t)}}\left(Z^{(t)}\right)}{\mathrm{Var}_{P^{(t+1)}}\left(Z^{(t)}\right)} \in [1-\zeta, 1+\zeta]$. This fact, together with some algebraic manipulations, lets us to reduce to the case that all $P^{(t)}$ are equal. It is also relatively straightforward to reduce to the case that $\langle P^{(t)}, Z^{(t)}\rangle = 0$ for all $t$, i.e., so that $\mathrm{Var}_{P^{(t)}}\left(Z^{(t)}\right) = \left\|Z^{(t)}\right\|_{P^{(t)}}^2$. We may further separate $\left\|Z^{(t)}\right\|_{P^{(t)}}^2 = \sum_{j=1}^{n} P^{(t)}(j) \cdot (Z^{(t)}(j))^2$ into its individual components $P^{(t)}(j) \cdot (Z^{(t)}(j))^2$, and treat each one separately, thus allowing us to reduce to a one-dimensional problem. Finally, we make one further reduction, which is to replace the finite differences $\mathrm{D}_h\,(\cdot)$ in Lemma 4.7 with *circular finite differences*, defined below:

**Definition 4.2** (Circular finite difference). Suppose $L = (L^{(0)}, \ldots, L^{(S-1)})$ is a sequence of vectors $L^{(t)} \in \mathbb{R}^n$. For integers $h \geq 0$, the *level-$h$ circular finite difference sequence* for the sequence $L$, denoted by $\mathrm{D}_h^\circ\, L$, is the sequence defined recursively as: $(\mathrm{D}_0^\circ\, L)^{(t)} = L^{(t)}$ for all $0 \leq t < S$, and

$$(\mathrm{D}_h^\circ\, L)^{(t)} = \begin{cases} \left(\mathrm{D}_{h-1}^\circ\, L\right)^{(t+1)} - \left(\mathrm{D}_{h-1}^\circ\, L\right)^{(t)} & : 0 \leq t \leq S-2 \\ \left(\mathrm{D}_{h-1}^\circ\, L\right)^{(1)} - \left(\mathrm{D}_{h-1}^\circ\, L\right)^{(T)} & : t = S-1. \end{cases} \tag{13}$$

Circular finite differences for a sequence $L^{(0)}, \ldots, L^{(S-1)}$ are defined similarly to finite differences (Definition 4.1) except that unlike for finite differences, where $(\mathrm{D}_h\, L)^{(S-h)}, \ldots, (\mathrm{D}_h\, L)^{(S-1)}$ are not defined, $(\mathrm{D}_h^\circ\, L)^{(S-h)}, \ldots, (\mathrm{D}_h^\circ\, L)^{(S-1)}$ are defined by "wrapping around" back to the beginning of the sequence. The above-described reductions, which are worked out in detail in Section C.2, allow us to reduce proving Lemma 4.7 to proving the following simpler lemma:

**Lemma 4.8.** *Suppose $\mu \in \mathbb{R}$, $\alpha > 0$, and $W^{(0)}, \ldots, W^{(S-1)} \in \mathbb{R}$ is a sequence of reals satisfying*

$$\sum_{t=0}^{S-1} \left((\mathrm{D}_2^\circ\, W)^{(t)}\right)^2 \leq \alpha \cdot \sum_{t=0}^{S-1} \left((\mathrm{D}_1^\circ\, W)^{(t)}\right)^2 + \mu. \tag{14}$$

*Then* $\sum_{t=0}^{S-1} \left((\mathrm{D}_1^\circ\, W)^{(t)}\right)^2 \leq \alpha \cdot \sum_{t=1}^{S-1} (W^{(t)})^2 + \mu/\alpha$.

To prove Lemma 4.8, we apply the discrete Fourier transform to both sides of (14) and use the Cauchy-Schwarz inequality in frequency domain. For a sequence $W^{(0)}, \ldots, W^{(S-1)} \in \mathbb{R}$, its (discrete) *Fourier transform* is the sequence $\widehat{W}^{(0)}, \ldots, \widehat{W}^{(S-1)}$ defined by $\widehat{W}^{(s)} = \sum_{t=0}^{S-1} W^{(t)} \cdot e^{-\frac{2\pi i s t}{S}}$. Below we prove Lemma 4.8 for the special case $\mu = 0$; we defer the general case to Section C.1.

*Proof of Lemma 4.8 for special case $\mu = 0$.* We have the following:

$$S \cdot \sum_{t=1}^{T} \left((\mathrm{D}_1^\circ\, W)^{(t)}\right)^2 = \sum_{s=1}^{T} \left|\widehat{\mathrm{D}_1^\circ\, W}^{(s)}\right|^2 = \sum_{s=1}^{T} \left|\widehat{W}^{(s)}(e^{2\pi i s/T} - 1)\right|^2 \leq \sqrt{\sum_{s=1}^{T} \left|\widehat{W}^{(s)}\right|^2} \sqrt{\sum_{s=1}^{T} \left|\widehat{W}^{(s)}\right|^2 \left|e^{2\pi i s/T} - 1\right|^4},$$

where the first equality uses Parseval's equality, the second uses Fact C.3 (in the appendix) for $h = 1$, and the inequality uses Cauchy-Schwarz. By Parseval's inequality and Fact C.3 for $h = 2$, the right-hand side of the above equals $S \cdot \sqrt{\sum_{t=1}^{T} (W^{(t)})^2} \cdot \sqrt{\sum_{t=1}^{T} \left((\mathrm{D}_2^\circ\, W)^{(t)}\right)^2}$, which, by assumption, is at most $S \cdot \sqrt{\sum_{t=1}^{T} (W^{(t)})^2} \cdot \sqrt{\alpha \cdot \sum_{t=1}^{T} \left((\mathrm{D}_1^\circ\, W)^{(t)}\right)^2}$. Rearranging terms completes the proof. $\quad\square$

## Acknowledgments and Disclosure of Funding

C.D. is Supported by NSF Awards CCF-1901292, DMS-2022448 and DMS-2134108, by a Simons Investigator Award, by the Simons Collaboration on the Theory of Algorithmic Fairness, by a DSTA grant, and by the DOE PhILMs project (No. DE-AC05-76RL01830). N.G. is supported by a Fannie & John Hertz Foundation Fellowship and an NSF Graduate Fellowship.

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
