$(\text{D}_1 Z)^{(t)} = (\text{D}_{h+1} \ell_i)^{(t)}$ and $(\text{D}_2 Z)^{(t)} = (\text{D}_{h+2} \ell_i)^{(t)}$, that the inductive hypothesis (12) holds for $h + 1$ implies that precondition 2 of Lemma 4.7 holds for appropriate $\alpha, \mu > 0$. Thus Lemma 4.7 implies that (12) holds for the value $h$, which completes the inductive step.

**On the proof of Lemma 4.7.** Finally we discuss the proof of Lemma 4.7. One technical challenge is the fact that the vectors $P^{(t)}$ are not constant functions of $t$, but rather change slowly (as constrained by being $\zeta$-consecutively close). The main tool for dealing with this difficulty is Lemma C.1, which shows that for a $\zeta$-consecutively close sequence $P^{(t)}$, for any vector $Z^{(t)}$, $\frac{\text{Var}_{P^{(t)}}\left(Z^{(t)}\right)}{\text{Var}_{P^{(t+1)}}\left(Z^{(t)}\right)} \in [1 - \zeta, 1 + \zeta]$. This fact, together with some algebraic manipulations, lets us to reduce to the case that all $P^{(t)}$ are equal. It is also relatively straightforward to reduce to the case that $\langle P^{(t)}, Z^{(t)} \rangle = 0$ for all $t$, i.e., so that $\text{Var}_{P^{(t)}}\left(Z^{(t)}\right) = \left\| Z^{(t)} \right\|_{P^{(t)}}^2$. We may further separate $\left\| Z^{(t)} \right\|_{P^{(t)}}^2 = \sum_{j=1}^{n} P^{(t)}(j) \cdot (Z^{(t)}(j))^2$ into its individual components $P^{(t)}(j) \cdot (Z^{(t)}(j))^2$, and treat each one separately, thus allowing us to reduce to a one-dimensional problem. Finally, we make one further reduction, which is to replace the finite differences $\text{D}_h (\cdot)$ in Lemma 4.7 with *circular finite differences*, defined below:

**Definition 4.2** (Circular finite difference). Suppose $L = (L^{(0)}, \ldots, L^{(S-1)})$ is a sequence of vectors $L^{(t)} \in \mathbb{R}^n$. For integers $h \ge 0$, the *level-$h$ circular finite difference sequence* for the sequence $L$, denoted by $\text{D}_h^\circ L$, is the sequence defined recursively as: $(\text{D}_0^\circ L)^{(t)} = L^{(t)}$ for all $0 \le t < S$, and

$$(\text{D}_h^\circ L)^{(t)} = \begin{cases} \left(\text{D}_{h-1}^\circ L\right)^{(t+1)} - \left(\text{D}_{h-1}^\circ L\right)^{(t)} & : 0 \le t \le S - 2 \\ \left(\text{D}_{h-1}^\circ L\right)^{(1)} - \left(\text{D}_{h-1}^\circ L\right)^{(T)} & : t = S - 1. \end{cases} \tag{13}$$

Circular finite differences for a sequence $L^{(0)}, \ldots, L^{(S-1)}$ are defined similarly to finite differences (Definition 4.1) except that unlike for finite differences, where $(\text{D}_h L)^{(S-h)}, \ldots, (\text{D}_h L)^{(S-1)}$ are not defined, $(\text{D}_h^\circ L)^{(S-h)}, \ldots, (\text{D}_h^\circ L)^{(S-1)}$ are defined by "wrapping around" back to the beginning of the sequence. The above-described reductions, which are worked out in detail in Section C.2, allow us to reduce proving Lemma 4.7 to proving the following simpler lemma:

**Lemma 4.8.** *Suppose $\mu \in \mathbb{R}$, $\alpha > 0$, and $W^{(0)}, \ldots, W^{(S-1)} \in \mathbb{R}$ is a sequence of reals satisfying*

$$\sum_{t=0}^{S-1} \left( (\text{D}_2^\circ W)^{(t)} \right)^2 \le \alpha \cdot \sum_{t=0}^{S-1} \left( (\text{D}_1^\circ W)^{(t)} \right)^2 + \mu. \tag{14}$$

*Then $\sum_{t=0}^{S-1} \left( (\text{D}_1^\circ W)^{(t)} \right)^2 \le \alpha \cdot \sum_{t=1}^{S-1} (W^{(t)})^2 + \mu/\alpha$.*

To prove Lemma 4.8, we apply the discrete Fourier transform to both sides of (14) and use the Cauchy-Schwarz inequality in frequency domain. For a sequence $W^{(0)}, \ldots, W^{(S-1)} \in \mathbb{R}$, its (discrete) *Fourier transform* is the sequence $\widehat{W}^{(0)}, \ldots, \widehat{W}^{(S-1)}$ defined by $\widehat{W}^{(s)} = \sum_{t=0}^{S-1} W^{(t)} \cdot e^{-\frac{2\pi i s t}{S}}$. Below we prove Lemma 4.8 for the special case $\mu = 0$; we defer the general case to Section C.1.

*Proof of Lemma 4.8 for special case $\mu = 0$.* We have the following:

$$S \cdot \sum_{t=1}^{T} \left( (\text{D}_1^\circ W)^{(t)} \right)^2 = \sum_{s=1}^{T} \left| \widehat{\text{D}_1^\circ W}^{(s)} \right|^2 = \sum_{s=1}^{T} \left| \widehat{W}^{(s)} (e^{2\pi i s/T} - 1) \right|^2 \le \sqrt{\sum_{s=1}^{T} \left| \widehat{W}^{(s)} \right|^2} \sqrt{\sum_{s=1}^{T} \left| \widehat{W}^{(s)} \right|^2 \left| e^{2\pi i s/T} - 1 \right|^4},$$

where the first equality uses Parseval's equality, the second uses Fact C.3 (in the appendix) for $h = 1$, and the inequality uses Cauchy-Schwarz. By Parseval's inequality and Fact C.3 for $h = 2$, the right-hand side of the above equals $S \cdot \sqrt{\sum_{t=1}^{T} (W^{(t)})^2} \cdot \sqrt{\sum_{t=1}^{T} \left( (\text{D}_2^\circ W)^{(t)} \right)^2}$, which, by assumption, is at most $S \cdot \sqrt{\sum_{t=1}^{T} (W^{(t)})^2} \cdot \sqrt{\alpha \cdot \sum_{t=1}^{T} \left( (\text{D}_1^\circ W)^{(t)} \right)^2}$. Rearranging terms completes the proof. $\square$

## Acknowledgments and Disclosure of Funding

C.D. is Supported by NSF Awards CCF-1901292, DMS-2022448 and DMS-2134108, by a Simons Investigator Award, by the Simons Collaboration on the Theory of Algorithmic Fairness, by a DSTA grant, and by the DOE PhILMs project (No. DE-AC05-76RL01830). N.G. is supported by a Fannie & John Hertz Foundation Fellowship and an NSF Graduate Fellowship.

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

# A Proofs for Section 4.1

In this section we prove Lemma 4.1. Throughout the section we use the notation of Lemma 4.1: in particular, we assume that any player $i \in [m]$ follows the Optimistic Hedge updates (1) with step size $\eta > 0$, for an arbitrary sequence of losses $\ell_i^{(1)}, \ldots, \ell_i^{(T)}$.

## A.1 Preliminary lemmas

The first few lemmas in this section pertain to vectors $P, Q \in \Delta^n$, for some $n \in \mathbb{N}$; note that such vectors $P, Q$ may be viewed as distributions on $[n]$. Let $P/Q \in \mathbb{R}^n$ denote the Radon-Nikodym derivative, i.e., the vector whose $j$th component is $P(j)/Q(j)$.

**Lemma A.1.** *If $\|P/Q\|_\infty \leq A$, then $\chi^2(P; Q) \leq A \cdot \chi^2(Q; P)$.*

*Proof.* The lemma is immediate from the definition of the $\chi^2$ divergence:

$$\chi^2(P; Q) = \sum_{j=1}^n \frac{(P(j) - Q(j))^2}{Q(j)} \leq A \cdot \sum_{j=1}^n \frac{(P(j) - Q(j))^2}{P(j)} = A \cdot \chi^2(Q; P).$$

$\square$

It is a standard fact (though one which we do not need in our proofs) that for all $P, Q \in \Delta^{n_i}$, $\mathrm{KL}(P; Q) \leq \chi^2(P; Q)$. The below lemma shows an inequality in the opposite direction when $\|P/Q\|_\infty, \|Q/P\|_\infty$ are bounded:

**Lemma A.2.** *There is a constant $C$ so that the following holds. Suppose that for $A \leq \frac{3}{2}$ we have $\|P/Q\|_\infty \leq A$ and $\|Q/P\|_\infty \leq A$. Then $(1/2 - C(A-1)) \cdot \chi^2(P; Q) \leq \mathrm{KL}(P; Q)$.*

*Proof.* There is a constant $C > 0$ so that for any $0 < \beta \leq 1/2$, for all $|x| \leq \beta$, we have

$$\log(1 + x) \geq x - (1/2 + C\beta)x^2.$$

Set $a = A - 1$, so that $|P(j)/Q(j) - 1| \leq a$ for all $j$ by assumption. Then for $C' = C + 1/2$, we have

$$\begin{aligned}
\mathrm{KL}(P; Q) &= \sum_j P(j) \log \frac{P(j)}{Q(j)} \\
&\geq \sum_j P(j) \cdot \left( \left( \frac{P(j)}{Q(j)} - 1 \right) - (1/2 + Ca) \left( \frac{P(j)}{Q(j)} - 1 \right)^2 \right) \\
&\geq \chi^2(P; Q) - (1/2 + Ca) \sum_j P(j) \cdot \frac{(P(j) - Q(j))^2}{Q(j)^2} \\
&\geq \chi^2(P; Q) - \frac{1/2 + Ca}{A} \chi^2(P; Q) \\
&\geq \frac{1/2 - aC}{1 + a} \cdot \chi^2(P; Q) \\
&\geq (1/2 - a \cdot (C + 1/2)) \cdot \chi^2(P; Q) \\
&= (1/2 - C'a) \cdot \chi^2(P; Q).
\end{aligned}$$

$\square$

The next lemma considers two vectors $x, x' \in \Delta^n$ which are related by a multiplicative weights-style update with loss vector $w \in \mathbb{R}^n$; the lemma relates $\chi^2(x'; x)$ to $\|w\|_x^2$.

**Lemma A.3.** *There is a constant $C > 0$ so that the following holds. Suppose that $w \in \mathbb{R}^n$, $\alpha > 0$, $\|w\|_\infty \leq \alpha/2 \leq 1/C$, and $x, x' \in \Delta^n$ satisfy, for each $j \in [n]$,*

$$x'(j) = \frac{x(j) \cdot \exp(w(j))}{\sum_{k \in [n]} x(k) \cdot \exp(w(k))}. \tag{15}$$

*Then*

$$(1 - C\alpha) \cdot \mathrm{Var}_x(w) \leq \chi^2(x'; x) \leq (1 + C\alpha) \mathrm{Var}_x(w).$$

*Proof.* Let $w' = w - \langle x, w \rangle \mathbf{1}$, where $\mathbf{1}$ denotes the all-1s vector. Note that $\mathrm{Var}_x(w) = \mathrm{Var}_x(w')$, and that if we replace $w$ with $w'$, (15) remains true. Moreover, $\|w'\|_\infty \le 2\|w\|_\infty \le \alpha$. Thus, by replacing $w$ with $w'$, we may assume from here on that $\langle w, x \rangle = 0$ and that $\|w\| \le \alpha$.

Note that

$$\chi^2(x'; x) = -1 + \sum_{i=1}^n x(i) \cdot (x'(i)/x(i))^2 = -1 + \mathbb{E}\left(\frac{\exp(W)}{\mathbb{E}\exp(W)}\right)^2,$$

where $W$ is a random variable that takes values $w(j)$ with probability $x(j)$. As long as $C$ is a sufficiently large constant, we have that, for all $z$ satisfying $|z| \le \alpha$,

$$1 + z + (1 - C\alpha)z^2/2 \le \exp(z) \le 1 + z + (1 + C\alpha)z^2/2. \tag{16}$$

Thus, for a sufficiently large constant $C_0'$, we have, for all $z$ satisfying $|z| \le \alpha$,

$$1 + 2z + (2 - C_0'\alpha)z^2 \le \exp(z)^2 \le 1 + 2z + (2 + C_0'\alpha)z^2. \tag{17}$$

Moreover, since $\mathbb{E}W = 0$, we have from (16) that $1 + (1 - C\alpha)\mathbb{E}W^2/2 \le \mathbb{E}\exp(W) \le 1 + (1 + C\alpha)\mathbb{E}W^2/2$. For a sufficiently large constant $C_1'$ it follows that

$$1 + (1 - C_1'\alpha)\mathbb{E}W^2 \le (\mathbb{E}\exp(W))^2 \le 1 + (1 + C_1'\alpha)\mathbb{E}W^2. \tag{18}$$

Combining (17) and (18) and again using the fact that $\mathbb{E}W = 0$, we get, for some sufficiently large constant $C''$, as long as $\alpha < 1/C_1'$,

$$
\begin{aligned}
(1 - C''\alpha)\mathbb{E}W^2 &\le -1 + \frac{1 + (2 - C_0'\alpha)\mathbb{E}W^2}{1 + (1 + C_1'\alpha)\mathbb{E}W^2} \\
&\le -1 + \frac{\mathbb{E}(\exp(W)^2)}{(\mathbb{E}\exp(W))^2} \\
&\le -1 + \frac{1 + (2 + C_0'\alpha)\mathbb{E}W^2}{1 + (1 - C_1'\alpha)\mathbb{E}W^2} \\
&\le (1 + C''\alpha)\mathbb{E}W^2.
\end{aligned}
$$

By the assumption that $\langle w, x \rangle = 0$, we have $\mathbb{E}W^2 = \mathrm{Var}_x(w)$, and thus the above gives the desired result. $\qquad\square$

We will need the following standard lemma:

**Lemma A.4** ([RS13a], Eq. (26)). *For any $n \in \mathbb{N}$, $\ell \in \mathbb{R}^n$, $y \in \Delta^n$, if it holds that $x = \arg\min_{x' \in \Delta^n}\langle x', \ell \rangle + \mathrm{KL}(x'; y)$, then for any $z \in \Delta^n$,*

$$\langle x - z, \ell \rangle \le \mathrm{KL}(z; y) - \mathrm{KL}(z; x) - \mathrm{KL}(x; y).$$

For $t \in [T]$, we define the vector $\tilde{x}_i^{(t)} \in \Delta^{n_i}$ by

$$\tilde{x}_i^{(t)}(j) := \frac{x_i^{(t)}(j) \cdot \exp(-\eta \cdot (\ell_i^{(t)}(j) - \ell_i^{(t-1)}(j)))}{\sum_{k \in [n_i]} x_i^{(t)}(k) \cdot \exp(-\eta \cdot (\ell_i^{(t)}(k) - \ell_i^{(t-1)}(k)))}. \tag{19}$$

Additionally define $\tilde{x}_i^{(0)} := (1/n_i, \ldots, 1/n_i)$ to be the uniform distribution over $[n_i]$.

The next lemma, Lemma A.5 is very similar to [RS13a, Lemma 3], and is indeed essentially shown in the course of the proof of that lemma. Note that no boundedness assumption is placed on the vectors $\ell_i^{(t)}$ in Lemma A.5. For completeness we provide a full proof of the lemma.

**Lemma A.5** (Refinement of Lemma 3, [RS13a]). *Suppose that any player $i \in [m]$ follows the Optimistic Hedge updates (1) with step size $\eta > 0$, for an arbitrary sequence of losses $\ell_i^{(1)}, \ldots, \ell_i^{(T)} \in \mathbb{R}^{n_i}$. For any vector $x^\star \in \Delta^{n_i}$, it holds that*

$$\sum_{t=1}^T \langle x_i^{(t)} - x^\star, \ell_i^{(t)} \rangle \le \frac{\log n_i}{\eta} + \sum_{t=1}^T \left\| x_i^{(t)} - \tilde{x}_i^{(t)} \right\|_{x_i^{(t)}}^\star \sqrt{\mathrm{Var}_{x_i^{(t)}}\left(\ell_i^{(t)} - \ell_i^{(t-1)}\right)} - \frac{1}{\eta}\sum_{t=1}^T \mathrm{KL}(\tilde{x}_i^{(t)}; x_i^{(t)}) - \frac{1}{\eta}\sum_{t=1}^T \mathrm{KL}(x_i^{(t)}; \tilde{x}_i^{(t-1)}).$$

$$\tag{20}$$

*Proof.* For any $x^\star \in \Delta^{n_i}$, it holds that

$$\langle x_i^{(t)} - x^\star, \ell_i^{(t)} \rangle = \langle x_i^{(t)} - \tilde{x}_i^{(t)}, \ell_i^{(t)} - \ell_i^{(t-1)} \rangle + \langle x_i^{(t)} - \tilde{x}_i^{(t)}, \ell_i^{(t-1)} \rangle + \langle \tilde{x}_i^{(t)} - x^\star, \ell_i^{(t)} \rangle. \quad (21)$$

For $t \in [T]$, set $c^{(t)} = \langle x_i^{(t)}, \ell_i^{(t)} - \ell_i^{(t-1)} \rangle$. Using the definition of the dual norm and the fact $\langle x_i^{(t)} - \tilde{x}_i^{(t)}, \mathbf{1} \rangle = 0$, we have

$$
\begin{aligned}
\langle x_i^{(t)} - \tilde{x}_i^{(t)}, \ell_i^{(t)} - \ell_i^{(t-1)} \rangle &= \langle x_i^{(t)} - \tilde{x}_i^{(t)}, \ell_i^{(t)} - \ell_i^{(t-1)} - c^{(t)}\mathbf{1} \rangle \\
&\leq \left\| x_i^{(t)} - \tilde{x}_i^{(t)} \right\|_{x_i^{(t)}}^\star \cdot \left\| \ell_i^{(t)} - \ell_i^{(t-1)} - c^{(t)}\mathbf{1} \right\|_{x_i^{(t)}} \\
&\leq \left\| x_i^{(t)} - \tilde{x}_i^{(t)} \right\|_{x_i^{(t)}}^\star \cdot \sqrt{\mathrm{Var}_{x_i^{(t)}} \left( \ell_i^{(t)} - \ell_i^{(t-1)} \right)}. \quad (22)
\end{aligned}
$$

It is immediate from the definitions of $\tilde{x}_i^{(t)}$ (in (19)) and $x_i^{(t)}$ (in (1)) that for $j \in [n_i]$,

$$x_i^{(t)}(j) = \frac{\tilde{x}_i^{(t-1)}(j) \cdot \exp(-\eta \cdot \ell_i^{(t-1)}(j))}{\sum_{k \in [n_i]} \tilde{x}_i^{(t-1)}(k) \cdot \exp(-\eta \cdot \ell_i^{(t-1)}(k))} = \arg\min_{x \in \Delta^{n_i}} \left\langle x, \eta \cdot \ell_i^{(t-1)} \right\rangle + \mathrm{KL}(x; \tilde{x}_i^{(t-1)}) \quad (23)$$

Using Lemma A.4 with $x = x_i^{(t)}, \ell = \eta \ell_i^{(t-1)}, y = \tilde{x}_i^{(t-1)}, z = \tilde{x}_i^{(t)}$, we obtain

$$\langle x_i^{(t)} - \tilde{x}_i^{(t)}, \ell_i^{(t-1)} \rangle \leq \frac{1}{\eta} \mathrm{KL}(\tilde{x}_i^{(t)}; \tilde{x}_i^{(t-1)}) - \frac{1}{\eta} \mathrm{KL}(\tilde{x}_i^{(t)}; x_i^{(t)}) - \frac{1}{\eta} \mathrm{KL}(x_i^{(t)}; \tilde{x}_i^{(t-1)}). \quad (24)$$

Next, we note that, again by (19) and (1), for $j \in [n_i]$,

$$\tilde{x}_i^{(t)}(j) = \frac{\tilde{x}_i^{(t-1)}(j) \cdot \exp(-\eta \cdot \ell_i^{(t)}(j))}{\sum_{k \in [n_i]} \tilde{x}_i^{(t-1)}(k) \cdot \exp(-\eta \cdot \ell_i^{(t)}(k))} = \arg\min_{x \in \Delta^{n_i}} \left\langle x, \eta \cdot \ell_i^{(t)} \right\rangle + \mathrm{KL}(x; \tilde{x}_i^{(t-1)}).$$

Using Lemma A.4 with $x = \tilde{x}_i^{(t)}, \ell = \eta \ell_i^{(t)}, y = \tilde{x}_i^{(t-1)}, z = x^\star$, we obtain

$$\langle \tilde{x}_i^{(t)} - x^\star, \ell_i^{(t)} \rangle \leq \frac{1}{\eta} \mathrm{KL}(x^\star; \tilde{x}_i^{(t-1)}) - \frac{1}{\eta} \mathrm{KL}(x^\star; \tilde{x}_i^{(t)}) - \frac{1}{\eta} \mathrm{KL}(\tilde{x}_i^{(t)}; \tilde{x}_i^{(t-1)}). \quad (25)$$

By (21), (22), (24), and (25), we have

$$
\begin{aligned}
\langle x_i^{(t)} - x^\star, \ell_i^{(t)} \rangle &\leq \left\| x_i^{(t)} - \tilde{x}_i^{(t)} \right\|_{x_i^{(t)}}^\star \cdot \sqrt{\mathrm{Var}_{x_i^{(t)}} \left( \ell_i^{(t)} - \ell_i^{(t-1)} \right)} \\
&\quad + \frac{1}{\eta} \mathrm{KL}(\tilde{x}_i^{(t)}; \tilde{x}_i^{(t-1)}) - \frac{1}{\eta} \mathrm{KL}(\tilde{x}_i^{(t)}; x_i^{(t)}) - \frac{1}{\eta} \mathrm{KL}(x_i^{(t)}; \tilde{x}_i^{(t-1)}) \\
&\quad + \frac{1}{\eta} \mathrm{KL}(x^\star; \tilde{x}_i^{(t-1)}) - \frac{1}{\eta} \mathrm{KL}(x^\star; \tilde{x}_i^{(t)}) - \frac{1}{\eta} \mathrm{KL}(\tilde{x}_i^{(t)}; \tilde{x}_i^{(t-1)}) \\
&= \left\| x_i^{(t)} - \tilde{x}_i^{(t)} \right\|_{x_i^{(t)}}^\star \cdot \sqrt{\mathrm{Var}_{x_i^{(t)}} \left( \ell_i^{(t)} - \ell_i^{(t-1)} \right)} + \frac{1}{\eta} \mathrm{KL}(x^\star; \tilde{x}_i^{(t-1)}) - \frac{1}{\eta} \mathrm{KL}(x^\star; \tilde{x}_i^{(t)}) \\
&\quad - \frac{1}{\eta} \mathrm{KL}(\tilde{x}_i^{(t)}; x_i^{(t)}) - \frac{1}{\eta} \mathrm{KL}(x_i^{(t)}; \tilde{x}_i^{(t-1)}). \quad (26)
\end{aligned}
$$

The statement of the lemma follows by summing (26) over $t \in [T]$ and using the fact that for any choice of $x^\star$, $\mathrm{KL}(x^\star; x_i^{(0)}) \leq \log n_i$. $\qquad\square$

## A.2 Proof of Lemma 4.1

Now we are ready to prove Lemma 4.1. For convenience we restate the lemma.

**Lemma 4.1** (restated). *There is a constant $C > 0$ so that the following holds. Suppose any player $i \in [m]$ follows the Optimistic Hedge updates (1) with step size $0 < \eta < 1/C$, for an arbitrary sequence of losses $\ell_i^{(1)}, \ldots, \ell_i^{(T)} \in [0, 1]^{n_i}$. Then for any vector $x^\star \in \Delta^{n_i}$, it holds that*

$$\sum_{t=1}^T \langle x_i^{(t)} - x^\star, \ell_i^{(t)} \rangle \leq \frac{\log n_i}{\eta} + \sum_{t=1}^T \left( \frac{\eta}{2} + C\eta^2 \right) \mathrm{Var}_{x_i^{(t)}} \left( \ell_i^{(t)} - \ell_i^{(t-1)} \right) - \sum_{t=1}^T \frac{(1 - C\eta)\eta}{2} \cdot \mathrm{Var}_{x_i^{(t)}} \left( \ell_i^{(t-1)} \right). \quad (27)$$

*Proof.* Lemma A.5 gives that, for any $x^\star \in \Delta^{n_i}$,

$$\sum_{t=1}^{T} \langle x_i^{(t)} - x^\star, \ell_i^{(t)} \rangle \leq \frac{\log n_i}{\eta} + \sum_{t=1}^{T} \left\| x_i^{(t)} - \tilde{x}_i^{(t)} \right\|_{x_i^{(t)}}^\star \sqrt{\mathrm{Var}_{x_i^{(t)}} \left( \ell_i^{(t)} - \ell_i^{(t-1)} \right)} - \frac{1}{\eta} \sum_{t=1}^{T} \mathrm{KL}(\tilde{x}_i^{(t)}; x_i^{(t)}) - \frac{1}{\eta} \sum_{t=1}^{T} \mathrm{KL}(x_i^{(t)}; \tilde{x}_i^{(t-1)}).$$
$$(28)$$

Note that for any vectors $x, x' \in \Delta^{n_i}$, if there is a vector $\ell \in \mathbb{R}^{n_i}$ so that for all $j \in [n_i]$, $x'(j) = \frac{x(j) \cdot \exp(\eta \cdot \ell(j))}{\sum_k x(j) \cdot \exp(\eta \cdot \ell(k))}$, we have that

$$\exp(-2\eta \|\ell\|_\infty) \leq \left\| \frac{x'}{x} \right\|_\infty \leq \exp(2\eta \|\ell\|_\infty).$$

Therefore, by (19) and (23), respectively, we obtain that, for $\eta \leq 1/4$,

$$\exp(-2\eta \|\ell_i^{(t)} - \ell_i^{(t-1)}\|_\infty) \leq \left\| \frac{\tilde{x}_i^{(t)}}{x_i^{(t)}} \right\|_\infty \leq \exp(2\eta \|\ell_i^{(t)} - \ell_i^{(t-1)}\|_\infty) \leq \exp(4\eta) \leq 1 + 8\eta$$

$$\exp(-2\eta \|\ell_i^{(t-1)}\|_\infty) \leq \left\| \frac{x_i^{(t)}}{\tilde{x}_i^{(t-1)}} \right\|_\infty \leq \exp(2\eta \|\ell_i^{(t-1)}\|_\infty) \leq \exp(2\eta) \leq 1 + 4\eta. \quad (29)$$

(Above we have also used that $\|\ell_i^{(t)}\|_\infty \leq 1$ for all $t$.) Thus, for $\eta \leq \frac{1}{16}$, we can apply Lemma A.2 and show, for a sufficiently large constant $C_0$,

$$\mathrm{KL}(\tilde{x}_i^{(t)}; x_i^{(t)}) \geq \chi^2(\tilde{x}_i^{(t)}; x_i^{(t)}) \cdot (1/2 - C_0\eta) \qquad (30)$$
$$\mathrm{KL}(x_i^{(t)}; \tilde{x}_i^{(t-1)}) \geq \chi^2(x_i^{(t)}; \tilde{x}_i^{(t-1)}) \cdot (1/2 - C_0\eta). \qquad (31)$$

Note also that for vectors $x, y$ we have that $\chi^2(x; y) = \left( \|x - y\|_y^\star \right)^2$. By Lemma A.3 and (19), we have that, for a sufficiently large constant $C_1$, as long as $\eta \leq 1/C_1$,

$$\left( \left\| x_i^{(t)} - \tilde{x}_i^{(t)} \right\|_{x_i^{(t)}}^\star \right)^2 = \chi^2(\tilde{x}_i^{(t)}; x_i^{(t)}) \leq (1 + C_1\eta)\eta^2 \cdot \mathrm{Var}_{x_i^{(t)}} \left( \ell_i^{(t)} - \ell_i^{(t-1)} \right) \qquad (32)$$

and

$$\chi^2(\tilde{x}_i^{(t)}; x_i^{(t)}) \geq (1 - C_1\eta)\eta^2 \cdot \mathrm{Var}_{x_i^{(t)}} \left( \ell_i^{(t)} - \ell_i^{(t-1)} \right). \qquad (33)$$

Next we lower bound $\chi^2(x_i^{(t)}; \tilde{x}_i^{(t-1)})$ as follows, where $C_2$ denotes a sufficiently large constant: as long as $\eta \leq 1/C_2$,

$$\chi^2(x_i^{(t)}; \tilde{x}_i^{(t-1)}) \geq \chi^2(\tilde{x}_i^{(t-1)}; x_i^{(t)}) \cdot \exp(-2\eta) \qquad (34)$$
$$\geq (1 - C_2\eta)\eta^2 \cdot \mathrm{Var}_{x_i^{(t)}} \left( \ell_i^{(t-1)} \right), \qquad (35)$$

where (34) follows from Lemma A.1 and (29), and (35) follows from Lemma A.3 and (23).

Combining (28), (30), (31), (32), (33), and (35) gives that for a sufficiently large constant $C$, as long as $\eta < 1/C$,

$$\sum_{t=1}^{T} \langle x_i^{(t)} - x^\star, \ell_i^{(t)} \rangle \leq \frac{\log n_i}{\eta} + \sum_{t=1}^{T} (\eta/2 + C\eta^2) \cdot \mathrm{Var}_{x_i^{(t)}} \left( \ell_i^{(t)} - \ell_i^{(t-1)} \right) - \frac{(1 - C\eta)\eta}{2} \cdot \mathrm{Var}_{x_i^{(t)}} \left( \ell_i^{(t-1)} \right),$$

as desired. □

## B Proofs for Section 4.3

In this section we give the full proof of Lemma 4.4. In Section B.1 we introduce some preliminaries. In Section B.2 we prove Lemma 4.5, the "boundedness chain rule" for finite differences. In Section B.4 we show how to use this lemma to prove Lemma 4.4.

## B.1 Additional preliminaries

In this section we introduce some additional notations and basic combinatorial lemmas. Definition B.1 introduces the *shift operator* $E_s$, which like the finite difference operator $D_h$, maps one sequence to another sequence.

**Definition B.1** (Shift operator). Suppose $L = (L^{(1)}, \ldots, L^{(T)})$ is a sequence of vectors $L^{(t)} \in \mathbb{R}^n$. For integers $s \geq 0$, the *s-shift sequence* for the sequence $L$, denoted by $E_s L$, is the sequence $E_s L = ((E_s L)^{(1)}, \ldots, (E_s L)^{(T-s)})$, defined by $(E_s L)^{(t)} = L^{(t+s)}$ for $1 \leq t \leq T - s$.

For sequences $L = (L^{(1)}, \ldots, L^{(T)})$ and $K = (K^{(1)}, \ldots, K^{(T)})$ of real numbers, we will denote the *product sequence* as $L \cdot K$ as the sequence of vectors $L \cdot K := (L^{(1)} K^{(1)}, \ldots, L^{(T)} K^{(T)})$. Lemmas B.1 and B.2 below are standard analogues of the product rule for finite differences. The (straightforward) proofs are provided for completeness.

**Lemma B.1** (Product rule; Eq. (2.55) of [GKP89]). *Suppose $L = (L^{(1)}, \ldots, L^{(T)})$ and $K = (K^{(1)}, \ldots, K^{(T)})$ are sequences of real numbers. Then the product sequence $L \cdot K$ satisfies*

$$D_1 (L \cdot K) = L \cdot D_1 K + D_1 L \cdot E_1 K.$$

*Proof.* We compute

$$
\begin{aligned}
D_1 (L \cdot K)^{(t)} &= L^{(t+1)} K^{(t+1)} - L^{(t)} K^{(t)} \\
&= L^{(t+1)} K^{(t+1)} - L^{(t)} K^{(t+1)} + L^{(t)} K^{(t+1)} - L^{(t)} K^{(t)} \\
&= (L \cdot D_1 K + D_1 L \cdot E_1 K)^{(t)}.
\end{aligned}
$$

$\square$

**Lemma B.2** (Multivariate product rule). *Suppose that $m \in \mathbb{N}$ and for $1 \leq i \leq m$, $L_i = (L_i^{(1)}, \ldots, L_i^{(T)})$ are sequences of real numbers. Then the product sequence $\prod_{i=1}^m L_i$ satisfies*

$$D_1 \prod_{i=1}^m L_i = \sum_{i=1}^m \left( \prod_{i' < i} L_{i'} \right) \cdot D_1 L_i \cdot \left( \prod_{i' > i} E_1 L_{i'} \right).$$

*Proof.* We compute

$$
\begin{aligned}
\left( D_1 \prod_{i=1}^m L_i \right)^{(t)} &= \prod_{i=1}^m L_i^{(t+1)} - \prod_{i=1}^m L_i^{(t)} \\
&= \sum_{i=1}^m \left( \prod_{i' \leq i} L_{i'}^{(t+1)} \prod_{i' > i} L_{i'}^{(t)} - \prod_{i' < i} L_{i'}^{(t+1)} \prod_{i' \geq i} L_{i'}^{(t)} \right) \\
&= \sum_{i=1}^m \left( \prod_{i' < i} L_{i'}^{(t+1)} \cdot \prod_{i' > i} L_{i'}^{(t)} \cdot \left( L_i^{(t+1)} - L_i^{(t)} \right) \right) \\
&= \left( \sum_{i=1}^m \left( \prod_{i' < i} L_{i'} \right) \cdot D_1 L_i \cdot \left( \prod_{i' > i} E_1 L_{i'} \right) \right)^{(t)}.
\end{aligned}
$$

$\square$

Lemma B.4 and Lemma B.3, which is used in the proof of the former, are used to bound certain sums with many terms in the proof of Lemma 4.5. To state Lemma B.3 we make one definition. For positive integers $k, m$ and any $h, C > 0$, define

$$R_{h,m,k,C} = \sum_{0 \leq n_1, \cdots, n_k \leq m} \left( \frac{\prod_{i=1}^k n_i^{n_i}}{h^{\sum_{i=1}^k n_i}} \right)^C,$$

where the sum is over integers $n_1, \ldots, n_k$ satisfying $0 \leq n_i \leq m$ for $i \in [k]$. In the definition of $R_{h,m,k,C}$, the quantity $0^0$ (which arises when some $n_i = 0$) is interpreted as 1.

**Lemma B.3.** *For any positive integers $k, m$ and any $h, C > 0$ so that $m \leq h/2$, $C \geq 2$, and $h \geq 8$, then*

$$R_{h,m,k,C} \leq \exp\left(\frac{2k}{h^C}\right).$$

*Proof of Lemma B.3.* We may rewrite $R_{h,m,k,C}$ and then upper bound it as follows:

$$R_{h,m,k,C} = \left(\sum_{j=0}^{m} \left(\frac{j}{h}\right)^{Cj}\right)^k$$

$$\leq \left(1 + \left(\frac{1}{h}\right)^C + (m-1)\max\left(\left(\frac{2}{h}\right)^{2C}, \left(\frac{m}{h}\right)^{mC}\right)\right)^k \tag{36}$$

$$\leq \left(1 + \left(\frac{1}{h}\right)^C + (h/2)\max\left(\left(\frac{2}{h}\right)^{2C}, \left(\frac{1}{2}\right)^{hC/2}\right)\right)^k$$

where (36) follows since $\left(\frac{i}{h}\right)^{Ci}$ is convex in $i$ for $i \geq 0$, and therefore, in the interval $[2, m] \subseteq [2, h/2]$, takes on maximal values at the endpoints. We see

$$(h/2)\left(\frac{2}{h}\right)^{2C} = \left(\frac{2}{h}\right)^{2C-1} \leq \left(\frac{1}{h}\right)^C$$

for $h \geq 8$ when $C \geq 2$. Also,

$$(h/2)\left(\frac{1}{2}\right)^{hC/2} \leq \left(\frac{1}{h}\right)^C$$

for $h \geq 8$ when $C \geq 2$. (This inequality is easily seen to be equivalent to the fact that $(C+1)\log h - \frac{Ch}{2} \leq 1$, which follows from the fact that $\log h - h/2 \leq 0$ for $h \geq 8$ and $3\log h - h \leq 1$ for $h \geq 8$.) Therefore,

$$R_{h,m,k,C} \leq \left(1 + \left(\frac{1}{h}\right)^C + (h/2)\max\left(\left(\frac{2}{h}\right)^{2C}, \left(\frac{1}{2}\right)^{hC/2}\right)\right)^k$$

$$\leq \left(1 + 2\left(\frac{1}{h}\right)^C\right)^k$$

$$\leq \exp\left(\frac{2k}{h^C}\right).$$

$\square$

**Lemma B.4.** *Fix integers $h \geq 0, k \geq 1$. For any function $\pi : [h] \to [k]$, define, for each $i \in [k]$, $h_i(\pi) = |\{q \in [h] | \pi(q) = i\}|$. Then, for any $C \geq 3$,*

$$\sum_{\pi:[h]\to[k]} \frac{\prod_{i=1}^{k} h_i(\pi)^{Ch_i(\pi)}}{h^{Ch}} \leq \max\left\{k^7, (hk+1) \cdot \exp\left(\frac{2k}{h^{C-1}}\right)\right\}. \tag{37}$$

*Proof.* In the case that $h \leq 7$, we simply use the fact that the number of functions $\pi : [h] \to [k]$ is $k^h \leq k^7$, and each term of the summation on the left-hand side of (37) is at most 1. In the remainder of the proof we may thus assume that $h \geq 8$.

For any tuple $(h_1, \cdots, h_k)$ of non-negative integers with $\sum_{i=1}^{k} h_i = h$, there are $\binom{h}{h_1, h_2, \cdots, h_k} \leq \frac{h^h}{\prod_i h_i^{h_i}}$ (see [CS04, Lemma 2.2] for a proof of this inequality) functions $\pi : [h] \to [k]$ such that

$h_i(\pi) = h_i$ for all $i \in [k]$. Combining these like terms,

$$\sum_{\pi:[h]\to[k]} \frac{\prod_i h_i(\pi)^{Ch_i(\pi)}}{h^{Ch}} \leq \sum_{\substack{h_1,\cdots,h_k\geq 0 \\ \sum h_i=h}} \frac{h^h}{\prod_i h_i^{h_i}} \cdot \left(\frac{\prod_i h_i^{h_i}}{h^h}\right)^C$$

$$\leq \sum_{\substack{h_1,\cdots,h_k\geq 0 \\ \sum h_i=h}} \left(\frac{\prod_i h_i^{h_i}}{h^h}\right)^{C-1}. \tag{38}$$

We evaluate this sum in 2 cases: whether or not $h_{\max} := \max_i\{h_i\}$ is greater than $h/2$. The contribution to this sum coming from terms with $h_{\max} \leq h/2$ is

$$\sum_{\substack{h_1,\cdots,h_k\geq 0 \\ h_1,\cdots,h_k\leq h/2 \\ \sum h_i=h}} \left(\frac{\prod_i h_i^{h_i}}{h^h}\right)^{C-1} \leq \sum_{\substack{h_1,\cdots,h_k\geq 0 \\ h_1,\cdots,h_k\leq h/2}} \left(\frac{\prod_i h_i^{h_i}}{h^{\sum h_i}}\right)^{C-1}$$

$$= R_{h,\lfloor h/2\rfloor,k,C-1}$$

$$\leq \exp\left(\frac{2k}{h^{C-1}}\right), \tag{39}$$

by Lemma B.3.

We next consider the case where $h_{\max} > h/2$. For a specific term $(h_1,\cdots,h_k)$ with $\max_i\{h_i\} > h/2$, we know there is a unique $M \in [k]$ such that $h_M = \max_i\{h_i\}$ since $\sum_{i=1}^k h_i = h$. So, we can represent the contribution to the sum from this case as

$$\sum_{M=1}^k \sum_{\substack{h_1,\cdots,h_k\geq 0 \\ h_M>h/2 \\ \sum h_i=h}} \left(\frac{\prod_i h_i^{h_i}}{h^h}\right)^{C-1} = k \sum_{\substack{h_1,\cdots,h_k\geq 0 \\ h_k>h/2 \\ \sum h_i=h}} \left(\frac{\prod_i h_i^{h_i}}{h^h}\right)^{C-1} \tag{40}$$

$$\leq k \sum_{d=0}^{\lfloor h/2\rfloor} \left(\frac{(h-d)^{h-d}}{h^{h-d}}\right)^{C-1} \sum_{\substack{h_1,\cdots,h_{k-1}\geq 0 \\ \sum h_i=d}} \left(\frac{\prod_i h_i^{h_i}}{h^d}\right)^{C-1} \tag{41}$$

$$\leq k \sum_{d=0}^{\lfloor h/2\rfloor} \sum_{\substack{h_1,\cdots,h_{k-1}\geq 0 \\ h_1,\cdots,h_{k-1}\leq d}} \left(\frac{\prod_i h_i^{h_i}}{h^{\sum h_i}}\right)^{C-1}$$

$$= k \sum_{d=0}^{\lfloor h/2\rfloor} R_{h,d,k-1,C-1}$$

$$\leq kh \cdot \exp\left(\frac{2k}{h^{C-1}}\right), \tag{42}$$

where (40) follows by symmetry, (41) follows by factoring out the contribution of $\left(\frac{h_k^{h_k}}{h^{h_k}}\right)^C$ and letting $d = h - h_k$, and (42) follows by Lemma B.3.

The statement of the lemma follows from (38), (39), and (42). $\qquad\square$

**Lemma B.5.** *For $n \in \mathbb{N}$, let $\xi_1,\ldots,\xi_n \geq 0$ such that $\xi_1 + \cdots + \xi_n = 1$. For each $j \in [n]$, define $\phi_j : \mathbb{R}^n \to \mathbb{R}$ to be the function*

$$\phi_j((z_1,\ldots,z_n)) = \frac{\xi_j \exp(z_j)}{\sum_{k=1}^n \xi_k \cdot \exp(z_k)}$$

*and let $P_{\phi_j}(z) = \sum_{\gamma \in \mathbb{Z}_{\geq 0}^n} a_{j,\gamma} \cdot z^\gamma$ denote the Taylor series of $\phi_j$. Then for any $j \in [n]$ and any integer $k \geq 1$,*

$$\sum_{\gamma \in \mathbb{Z}_{\geq 0}^n: \, |\gamma| = k} |a_{j,\gamma}| \leq \xi_j e^{k+1}.$$

*Proof.* Note that, for each $j \in [n]$,

$$a_{j,\gamma} = \frac{1}{\gamma_1! \gamma_2! \cdots \gamma_n!} \cdot \frac{\partial^k \phi_j(0)}{\partial z_1^{\gamma_1} \partial z_2^{\gamma_2} \cdots z_n^{\gamma_n}},$$

and so

$$\sum_{\gamma \in \mathbb{Z}_{\geq 0}^n: \, |\gamma| = k} |a_{j,\gamma}| = \sum_{\gamma \in \mathbb{Z}_{\geq 0}^n: \, |\gamma| = k} \frac{1}{\gamma_1! \gamma_2! \cdots \gamma_n!} \cdot \left| \frac{\partial^k \phi_j(0)}{\partial z_1^{\gamma_1} \partial z_2^{\gamma_2} \cdots z_n^{\gamma_n}} \right|$$

$$= \frac{1}{k!} \sum_{\gamma \in \mathbb{Z}_{\geq 0}^n: \, |\gamma| = k} \frac{k!}{\gamma_1! \gamma_2! \cdots \gamma_n!} \cdot \left| \frac{\partial^k \phi_j(0)}{\partial z_1^{\gamma_1} \partial z_2^{\gamma_2} \cdots z_n^{\gamma_n}} \right|$$

$$= \frac{1}{k!} \sum_{t \in [n]^k} \left| \frac{\partial^k \phi_j(0)}{\partial z_{t_1} \partial z_{t_2} \cdots \partial z_{t_k}} \right|.$$

It is straightforward to see that the following equalities hold for any $i \in [n]$, $i \neq j$:

$$\frac{\partial \phi_j}{\partial z_j} = \phi_j(1 - \phi_j)$$

$$\frac{\partial \phi_j}{\partial z_i} = -\phi_i \phi_j$$

$$\frac{\partial(1 - \phi_j)}{\partial z_j} = -\phi_j(1 - \phi_j)$$

$$\frac{\partial(1 - \phi_j)}{\partial z_i} = \phi_i \phi_j$$

We claim that for any $(t_1, \ldots, t_k) \in [n]^k$, we can express $\frac{\partial^k \phi_j}{\partial z_{t_1} \cdots \partial z_{t_k}}$ as a polynomial in $\phi_1, \cdots, \phi_n, (1 - \phi_1), \cdots, (1 - \phi_n)$ comprised of $k!$ monomials each of degree $k + 1$. We verify this by induction, first noting that after taking zero derivatives, the function $\phi_j$ is a degree-1 monomial. Assume that for some sequence $b_1, \ldots, b_{(\ell-1)!} \in \{0, 1\}$, we can express

$$\frac{\partial^{\ell-1} \phi_j}{\partial z_{t_1} \cdots \partial z_{t_{\ell-1}}} = \sum_{f=1}^{(\ell-1)!} (-1)^{b_f} \prod_{d=0}^{\ell-1} m_{f,d}$$

where each $m_{f,d} \in \{\phi_1, \cdots, \phi_n, (1 - \phi_1), \cdots, (1 - \phi_n)\}$. We see that for each $f$, there is some sequence of bits $b_{f,0}, \ldots, b_{f,\ell-1} \in \{0, 1\}$ so that

$$\frac{\partial}{\partial z_{t_\ell}} \prod_{d=0}^{\ell-1} m_{f,d} = \sum_{d=0}^{\ell-1} (-1)^{b_{f,d}} \cdot m_{f,0} \cdots m'_{f,d} \cdots m_{f,d,\ell} \tag{43}$$

where we define, for each $0 \leq d \leq \ell - 1$,

$$m'_{f,d} \text{ and } m_{f,d,\ell} = \begin{cases} m_{f,d} \text{ and } \phi_{t_\ell} & \text{if } m_{f,d} = \phi_i \text{ with } i \neq t_\ell \\ m_{f,d} \text{ and } (1 - \phi_{t_\ell}) & \text{if } m_{f,d} = \phi_{t_\ell} \\ (1 - m_{f,d}) \text{ and } \phi_{t_\ell} & \text{if } m_{f,d} = 1 - \phi_i \text{ with } i \neq t_\ell \\ (1 - m_{f,d}) \text{ and } (1 - \phi_{t_\ell}) & \text{if } m_{f,d} = 1 - \phi_{t_\ell}. \end{cases}$$

Thus, $\frac{\partial^\ell \phi_j}{\partial z_{t_1} \cdots \partial z_{t_\ell}}$ can be expressed as a sum of $\ell!$ monomials of degree $(\ell + 1)$, completing the inductive step.

This inductive argument also demonstrates a bijection between the $k!$ monomials of $\frac{\partial^k \phi_j}{\partial z_{t_1} \cdots \partial z_{t_k}}$ and a combinatorial structure that we call *factorial trees*. Formally, we define a factorial tree to be a directed graph on vertices $\{0, 1, \cdots, k\}$ such that each vertex $i \neq 0$ has a single incoming edge from one of the vertices in $[0, i-1]$. (For a non-negative integer $i$, we write $[0, i] := \{0, 1, \ldots, i\}$.) For a factorial tree $f$, let $p_f(\ell) \in [0, \ell-1]$ denote the parent of a vertex $\ell$. A particular factorial tree $f$ represents the monomial that was generated by choosing the $p_f(\ell)^{\text{th}}$ term in (43) for derivation when taking the derivative $\frac{\partial}{\partial z_{t_\ell}}$, for each $\ell \in [k]$. (See Figure 1 for an example.)

$$\phi_j \xrightarrow{\ \partial/\partial z_i\ } -\phi_j\phi_i \xrightarrow{\ \partial/\partial z_j\ } -\phi_j\phi_i(1 - \phi_j) \xrightarrow{\ \partial/\partial z_k\ } -\phi_j\phi_i\phi_j\phi_k$$

Figure 1: A monomial $-\phi_j\phi_i\phi_j\phi_k$ of $\frac{\partial^3 \phi_j}{\partial z_i \partial z_j \partial z_k}$ and its corresponding factorial tree

Each of the $k!$ monomials comprising $\frac{\partial^k \phi_j}{\partial z_{t_1} \cdots \partial z_{t_k}}$ is a product of $k + 1$ terms corresponding to indices $j, t_1, \cdots, t_k$ (i.e., the first term in the product is either $\phi_j$ or $1 - \phi_j$, the second term is either $\phi_{t_1}$ or $1 - \phi_{t_1}$, and so on). We say that a term corresponding to index $i \in [n]$ is *perturbed* if it is $(1 - \phi_i)$ (as opposed to $\phi_i$). From our construction, we see that the $\ell^{\text{th}}$ term is perturbed if $t_\ell = t_{p_f(\ell)}$ and there is no $\ell'$ such that $p_f(\ell') = \ell$. That is, $\ell$ is a leaf in the corresponding factorial tree $f$ and the parent of $\ell$ corresponds to the same index as $\ell$. One can think of $t_1, \cdots, t_k$ as a coloring of all the vertices of the factorial tree with $n$ colors, except the root of the tree (vertex 0) which has fixed color $j$. Then, we can say the $\ell^{\text{th}}$ term is perturbed if and only if $\ell$ is a leaf with the same color as its parent. We call such a leaf a *petal*. For $t \in [n]$, we let $P_{f,t} \subseteq [k]$ be the set of petals on tree $f$ with color $t$, $L_f \subseteq [k]$ be the set of leaves of tree $f$, and $B_f = [k] \setminus L_f$ be the set of all non-leaves other than the fixed-color root. Therefore,

$$\sum_{\gamma \in \mathbb{Z}_{\geq 0}^n : |\gamma| = k} |a_{j,\gamma}| = \frac{1}{k!} \sum_{t \in [n]^k} \left| \frac{\partial^k \phi_j(0)}{\partial z_{t_1} \cdots \partial z_{t_k}} \right|$$

$$\leq \frac{1}{k!} \sum_{t \in [n]^k} \sum_f \prod_{\ell=0}^{k} (\phi_{t_\ell}(0) \cdot \mathbb{1}[\ell \notin P_{f,t}] + (1 - \phi_{t_\ell}(0)) \cdot \mathbb{1}[\ell \in P_{f,t}])$$

(where we let $t_0 = j$ for notational convenience)

$$= \frac{1}{k!} \sum_{t \in [n]^k} \sum_f \prod_{\ell=0}^{k} (\xi_{t_\ell} \cdot \mathbb{1}[\ell \notin P_{f,t}] + (1 - \xi_{t_\ell}) \cdot \mathbb{1}[\ell \in P_{f,t}])$$

$$= \frac{1}{k!} \sum_f \sum_{t_{B_f} \in [n]^{B_f}} \sum_{t_{L_f} \in [n]^{L_f}} \prod_{\ell=0}^{k} (\xi_{t_\ell} \cdot \mathbb{1}[\ell \notin P_{f,t}] + (1 - \xi_{t_\ell}) \cdot \mathbb{1}[\ell \in P_{f,t}]),$$

where in the last step we decompose, for each factorial tree $f$, $t \in [n]^k$ into the tuple of indices $t_{B_f} \in [n]^{B_f}$ corresponding to the non-leaves $B_f$, and the tuple of indices $t_{L_f} \in [n]^{L_f}$ corresponding to the leaves $L_f$.

We note that, fixing tree $f$ and the colors of all non-leaves $t_B$,

$$\sum_{t_{L_f} \in [n]^{L_f}} \prod_{\ell \in L_f} (\xi_{t_\ell} \cdot \mathbb{1}[\ell \notin P_{f,t}] + (1 - \xi_{t_\ell}) \cdot \mathbb{1}[\ell \in P_{f,t}])$$

$$= \prod_{\ell \in L_f} \left( \sum_{t_\ell \in [n]} \xi_{t_\ell} \cdot \mathbb{1}[t_\ell \neq t_{p_f(\ell)}] + (1 - \xi_{t_\ell}) \cdot \mathbb{1}[t_\ell = t_{p_f(\ell)}] \right)$$

$$= \prod_{\ell \in L_f} \left( 2 - 2\xi_{t_{p_f(\ell)}} \right)$$

$$\leq 2^{|L_f|}$$

And so,

$$\frac{1}{k!} \sum_f \sum_{t_{B_f} \in [n]^{B_f}} \sum_{t_{L_f} \in [n]^{L_f}} \prod_{\ell=0}^{k} (\xi_{t_\ell} \cdot \mathbb{1}[\ell \notin P_{f,t}] + (1 - \xi_{t_\ell}) \cdot \mathbb{1}[\ell \in P_{f,t}])$$

$$\leq \frac{1}{k!} \sum_f 2^{|L_f|} \sum_{t_{B_f} \in [n]^{B_f}} \prod_{\ell \in B_f \cup \{0\}} (\xi_{t_\ell} \cdot \mathbb{1}[\ell \notin P_{f,t}] + (1 - \xi_{t_\ell}) \cdot \mathbb{1}[\ell \in P_{f,t}])$$

$$= \frac{1}{k!} \sum_f 2^{|L_f|} \sum_{t_{B_f} \in [n]^{B_f}} \prod_{\ell \in B_f \cup \{0\}} \xi_{t_\ell}$$

(as no non-leaf can ever be a petal)

$$= \frac{\xi_j}{k!} \sum_f 2^{|L_f|} \prod_{\ell \in B_f} \left( \sum_{t_\ell \in [n]} \xi_{t_\ell} \right)$$

$$= \frac{\xi_j}{k!} \sum_f 2^{|L_f|} = \xi_j \mathbb{E}_{f \sim \mathcal{U}(\mathcal{F})} \left[ 2^{|L_f|} \right]$$

where $\mathcal{F}$ is the set of all factorial trees and $\mathcal{U}(\mathcal{F})$ is the uniform distribution over $\mathcal{F}$. For a specific vertex $\ell \in [0, k]$, we note that $\ell \in L_f$ if and only if it is not the parent of any vertex $\ell + 1, \cdots, k$. So,

$$\Pr_{f \sim \mathcal{U}(\mathcal{F})} [\ell \in L_f] = \prod_{i=\ell+1}^{k} \frac{i-1}{i} = \frac{\ell}{k} \tag{44}$$

We will show via induction that, for any vertex set $S \subseteq [0, k]$

$$\Pr_{f \sim \mathcal{U}(\mathcal{F})} [S \subseteq L_f] \leq \prod_{\ell \in S} \frac{\ell}{k} \tag{45}$$

Having established the base case for every $S$ with $|S| = 1$, we assume (45) holds for all $S$ with $|S| < s$. For any set of $s$ vertices $V$, consider an arbitrary partition of $V$ into two sets $S \cup T = V$ with $|S|, |T| < s$. We see

$$\Pr_{f \sim \mathcal{U}(\mathcal{F})}[V \subseteq L_f] = \prod_{c=1}^{k} \Pr[p_f(c) \notin V]$$

$$= \prod_{c=1}^{k} \Pr[p_f(c) \notin S] \Pr[p_f(c) \notin T | p_f(c) \notin S]$$

$$\leq \prod_{c=1}^{k} \Pr[p_f(c) \notin S] \Pr[p_f(c) \notin T]$$

$$= \Pr[S \subseteq L_f] \Pr[T \subseteq L_f]$$

$$\leq \prod_{\ell \in V} \frac{\ell}{k}$$

by the inductive hypothesis, as desired. Thus, $\Pr[|L_f| \geq s] \leq \sum_{S:|S|=s} \Pr[S \subseteq L_f]$ is at most the $s^{\text{th}}$ coefficient of the polynomial

$$R(x) = \prod_{\ell=0}^{k} \left(1 + \frac{\ell}{k} x\right)$$

and so

$$\mathbb{E}_{f \sim \mathcal{U}(\mathcal{F})} \left[2^{|L_f|}\right] \leq \sum_{s=0}^{k} 2^s \Pr[|L_f| \geq s]$$

$$\leq R(2)$$

$$\leq e^{\sum_{\ell=0}^{k} 2\ell/k} = e^{k+1}$$

and

$$\sum_{\gamma \in \mathbb{Z}_{\geq 0}^n : |\gamma|=k} |a_{j,\gamma}| \leq \xi_j e^{k+1},$$

as desired. $\qquad \square$

**Lemma B.6.** *Let $\phi_1, \cdots, \phi_m$ be softmax-type functions. That is, for each $\phi_i$, there is some $j_i \in [n]$ and indices $\xi_{i1}, \ldots, \xi_{in}$ such that*

$$\phi_i((z_1, \ldots, z_n)) = \frac{\exp(z_{j_i})}{\sum_{k=1}^{n} \xi_{ik} \cdot \exp(z_k)}$$

*where $\xi_{i1} + \cdots + \xi_{in} = 1$ for all $i$. Let $P(z) = \sum_{\gamma \in \mathbb{Z}_{\geq 0}^n} a_\gamma z^\gamma$ denote the Taylor series of $\prod_i \phi_i$. Then for any integer $k$,*

$$\sum_{\gamma \in \mathbb{Z}_{\geq 0}^n : |\gamma|=k} |a_\gamma| \leq (e^3 m)^k.$$

*Proof.* Letting $P_i(z) = \sum_{\gamma \in \mathbb{Z}_{\geq 0}^n} a_{i,\gamma} z^\gamma$ denote the Taylor series of $\phi_i$ for all $i$, we have $P(z) = \prod_i P_i(z)$ and therefore

$$\sum_{\gamma \in \mathbb{Z}_{\geq 0}^n : |\gamma|=k} |a_\gamma| \leq \sum_{\substack{k_1, \cdots, k_m \in \mathbb{Z}_{\geq 0} \\ \sum k_i = k}} \prod_i \sum_{\gamma \in \mathbb{Z}_{\geq 0}^n : |\gamma|=k_i} |a_{i,\gamma}|$$

We have that $\sum_{|\gamma|=k_i} |a_{i,\gamma}| \leq e^{2k_i}$ for all $k_i$ since, for $k_i = 0$, $a_{i,0} = \phi_i(0) = 1$, and for $k_i \geq 1$,

$$\sum_{|\gamma|=k_i} |a_{i,\gamma}| \leq \frac{\xi_{ij}}{\xi_{ij}} e^{k_i+1} \leq e^{2k_i} \tag{46}$$

from Lemma B.5. Note that the softmax-type functions discussed in Lemma B.5 have a $\xi_{ij}$ term in the numerator, while those discussed here do not. This accounts for the extra $\xi_{ij}$ term that appears in equation (46). Thus,

$$\sum_{\substack{k_1,\cdots,k_m\in\mathbb{Z}_{\geq 0} \\ \sum k_i=k}} \prod_i \sum_{\gamma\in\mathbb{Z}_{\geq 0}^n:\,|\gamma|=k_i} |a_{i,\gamma}| \leq \sum_{\substack{k_1,\cdots,k_m\in\mathbb{Z}_{\geq 0} \\ \sum k_i=k}} e^{2k}$$

$$= e^{2k}\binom{m+k-1}{k}$$

$$\leq e^{2k}\left(\frac{e(m+k-1)}{k}\right)^k$$

$$= (e^3 m)^k$$

as desired. $\qquad\square$

**Lemma B.7.** *Let $\phi((z_1,\ldots,z_n)) = \frac{\exp(z_j)}{\sum_{k=1}^n \xi_k \exp(z_k)}$ be any softmax-type function. Then the radius of convergence of the Taylor series of $\phi$ at the origin is at least 1.*

*Proof.* For a complex number $z$, write $\Re(z), \Im(z)$ to denote the real and imaginary parts, respectively, of $z$. Note that for any $\zeta_1,\ldots,\zeta_n\in\mathbb{C}$ with $|\zeta_k|\leq \pi/3$ for all $k\in[n]$, we have

$$\Re(\exp(\zeta_k)) \geq \cos(\pi/3)\cdot\exp(-\pi/3) > 1/10,$$

and thus $|\sum_{k=1}^n \xi_k\cdot\exp(\zeta_k)| \geq 1/10$. Moreover, for any such point $\zeta = (\zeta_1,\ldots,\zeta_n)$, it holds that $|\exp(\zeta_j)| \leq \exp(\pi/3) < 3$. It then follows that for such $\zeta$ we have $|\phi(z)|\leq 30$. In particular, $\phi$ is holomorphic on the region $\{\zeta : |\zeta_k|\leq \pi/3\ \forall k\in[n]\}$.

Fix any $\gamma\in\mathbb{Z}_{\geq 0}^n$, and let $k=|\gamma|$. By the multivariate version of Cauchy's integral formula,

$$\left|\frac{d^\gamma}{dz^\gamma}\phi(z)\right| = \left|\frac{\gamma!}{(2\pi i)^n}\int_{|\zeta_1-z_1|=\pi/3}\cdots\int_{|\zeta_n-z_n|=\pi/3}\frac{\phi(\zeta_1,\ldots,\zeta_n)}{(\zeta_1-z_1)^{\gamma_1+1}\cdots(\zeta_n-z_n)^{\gamma_n+1}}d\zeta_1\cdots d\zeta_n\right|$$

$$\leq \frac{30\gamma!}{(\pi/3)^{k+n}} \leq \frac{30\gamma!}{(\pi/3)^k}.$$

The power series of $\phi$ at $\mathbf{0}$ is defined as $P_\phi(z) = \sum_{\gamma\in\mathbb{Z}_{\geq 0}^n} a_\gamma\cdot z^\gamma$, where $a_\gamma = \frac{1}{\gamma!}\frac{d^\gamma}{dz^\gamma}\phi(\mathbf{0})$. For any $\gamma\in\mathbb{Z}_{\geq 0}^n$ with $k=|\gamma|$, we have $|a_\gamma|^{1/k} \leq (30/(\pi/3)^k)^{1/k} = (30)^{1/k}\cdot 3/\pi$, which tends to $3/\pi < 1$ as $k\to\infty$. Thus, by the (multivariate version of the) Cauchy-Hadamard theorem, the radius of convergence of the power series of $\phi$ at $\mathbf{0}$ is at least $\pi/3 \geq 1$. $\qquad\square$

## B.2 Proof of Lemma 4.6

In this section prove Lemma 4.6, which, as explained in Section 4.3, is an important ingredient in the proof of Lemma 4.5. The detailed version of Lemma 4.6 is presented below; it includes several claims which are omitted for simplicity in the abbreviated version in Section 4.3.

**Lemma 4.6** (Detailed). *Fix any integer $h\geq 0$, a multi-index $\gamma\in\mathbb{Z}_{\geq 0}^n$ and set $k=|\gamma|$. For each of the $k^h$ functions $\pi : [h]\to[k]$, and for each $r\in[k]$, there are integers $h'_{\pi,r}\in\{0,1,\ldots,h\}$, $t'_{\pi,r}\geq 0$, and $j'_{\pi,r}\in[n]$, so that the following holds. For any sequence $L^{(1)},\ldots,L^{(T)}\in\mathbb{R}^n$ of vectors, it holds that*

$$\mathrm{D}_h\, L^\gamma = \sum_{\pi:[h]\to[k]}\prod_{r=1}^k \mathrm{E}_{t'_{\pi,r}}\, \mathrm{D}_{h'_{\pi,r}}\,(L(j'_{\pi,r})). \tag{47}$$

*Moreover, the following properties hold:*

1. *For each $\pi$ and $r\in[k]$, $h'_{\pi,r} = |\{q\in[h] : \pi(q)=r\}|$. In particular, $\sum_{r=1}^k h'_{\pi,r} = h$.*

2. *For each $\pi$ and $r\in[k]$, it holds that $0\leq t'_{\pi,r}+h'_{\pi,r}\leq h$.*

3. *For each $\pi$, $r \in [k]$, and $j \in [n]$, $\gamma_j = |\{r \in [k] : j'_{\pi,r} = j\}|$.*

*Proof of Lemma 4.6.* We use induction on $h$. First note that in the case $h = 0$ and for any $k \geq 0$, we have that $(D_h L^\gamma)^{(t)} = (L^{(t)})^\gamma$, and so for the unique function $\pi : \emptyset \to [k]$, for all $r \in [k]$, we may take $t'_{\pi,r} = 0$, $h'_{\pi,r} = 0$, and ensure that for each $j \in [n]$ there are $\gamma_j$ values of $r$ so that $j'_{\pi,r} = j$.

Now fix any integer $h > 0$, and suppose the statement of the claim holds for all $h' < h$. We have that

$$D_h L^\gamma$$
$$= D_1 D_{h-1} L^\gamma$$

$$= D_1 \sum_{\pi:[h-1]\to[k]} \prod_{r=1}^{k} E_{t'_{\pi,r}} D_{h'_{\pi,r}} L(j'_{\pi,r})$$

$$= \sum_{\pi:[h-1]\to[k]} \sum_{r=1}^{k} D_1 E_{t'_{\pi,r}} D_{h'_{\pi,r}} L(j'_{\pi,r}) \cdot \prod_{r'=1}^{r-1} E_{t'_{\pi,r'}} D_{h'_{\pi,r'}} L(j'_{\pi,r'}) \cdot \prod_{r'=r+1}^{k} E_{t'_{\pi,r'}+1} D_{h'_{\pi,r'}} L(j'_{\pi,r'})$$

(48)

$$= \sum_{\pi:[h-1]\to[k]} \sum_{r=1}^{k} E_{t'_{\pi,r}} D_{h'_{\pi,r}+1} L(j'_{\pi,r}) \cdot \prod_{r'=1}^{r-1} E_{t'_{\pi,r'}} D_{h'_{\pi,r'}} L(j'_{\pi,r'}) \cdot \prod_{r'=r+1}^{k} E_{t'_{\pi,r'}+1} D_{h'_{\pi,r'}} L(j'_{\pi,r'}).$$

(49)

where (48) uses Lemma B.2 and (49) uses the commutativity of $E_{t'}$ and $D_1$. For each $\pi : [h-1] \to [k]$, we construct $k$ functions $\pi_1, \ldots, \pi_k : [h] \to [k]$, defined by $\pi_r(q) = \pi(q)$ for $q < h$, and $\pi_r(h) = r$ for $r \in [k]$. Next, for $r, r' \in [k]$, we define the quantities $h'_{\pi_r,r'}, t'_{\pi_r,r'}, j'_{\pi_r,r'}$ as follows:

- Set $h'_{\pi_r,r'} = h'_{\pi,r'}$ if $r \neq r'$, and $h'_{\pi_r,r} = h'_{\pi,r} + 1$.
- Set $t'_{\pi_r,r'} = t'_{\pi,r'}$ if $r' \leq r$, and $t'_{\pi_r,r'} = t'_{\pi,r'} + 1$ if $r' > r$.
- Set $j'_{\pi_r,r'} = j'_{\pi,r'}$.

By (49) and the above definitions, we have

$$D_h L^\gamma = \sum_{\pi:[h]\to[k]} \prod_{r'=1}^{k} E_{t'_{\pi,r}} D_{h'_{\pi,r}} L(j'_{\pi,r}),$$

thus verifying (9) for the value $h$.

Finally we verify that items 1 through 3 in the lemma statement hold. The definition of $h'_{\pi_r,r'}$ above together with the inductive hypothesis ensures that for all $r, r' \in [k]$, $h'_{\pi_r,r'} = |\{q \in [h] : \pi_r(q) = r'\}|$, thus verifying item 1 of the lemma statement. Since $h'_{\pi_r,r'} + t'_{\pi_r,r'} \leq h'_{\pi,r} + t'_{\pi,r} + 1$ for all $r, r'$, it follows from the inductive hypothesis that $0 \leq h'_{\pi_r,r'} + t'_{\pi_r,r'} \leq h$; this verifies item 2. Finally, note that for any $j \in [n]$ and $r \in [k]$, $\{r' \in [k] : j'_{\pi,r'} = j\} = \{r' \in [k] : j'_{\pi_r,r'} = j\}$, and thus item 3 follows from the inductive hypothesis. $\qquad\square$

### B.3 Proof of Lemma 4.5

In this section we prove Lemma 4.5. To introduce the detailed version of the lemma we need the following definition. Suppose $\phi : \mathbb{R}^n \to \mathbb{R}$ is a real-valued function that is real-analytic in a neighborhood of the origin. For real numbers $Q, R > 0$, we say that $\phi$ is $(Q, R)$-*bounded* if the Taylor series of $\phi$ at $\mathbf{0}$, denoted $P_\phi(z_1, \ldots, z_n) = \sum_{\gamma \in \mathbb{Z}_{\geq 0}^n} a_\gamma z^\gamma$, satisfies, for each integer $k \geq 0$, $\sum_{\gamma \in \mathbb{Z}_{\geq 0}^n : |\gamma|=k} |a_\gamma| \leq Q \cdot R^k$. In the statement of Lemma 4.5 below, the quantity $0^0$ is interpreted as 1 (in particular, $(h')^{B_0 h'} = 1$ for $h' = 0$).

**Lemma 4.5** ("Boundedness chain rule" for finite differences; detailed). *Suppose that $h, n \in \mathbb{N}$, $\phi : \mathbb{R}^n \to \mathbb{R}$ is a $(Q, R)$-bounded function so that the radius of convergence of its power series at*

**0** *is at least 1, and* $L = (L^{(1)}, \ldots, L^{(T)}) \in \mathbb{R}^n$ *is a sequence of vectors satisfying* $\|L^{(t)}\|_\infty \le 1$ *for* $t \in [T]$. *Suppose for some* $\alpha \in (0, 1)$, *for each* $0 \le h' \le h$ *and* $t \in [T - h']$, *it holds that* $\|\operatorname{D}_{h'} L^{(t)}\|_\infty \le \frac{1}{B_1} \cdot \alpha^{h'} \cdot (h')^{B_0 h'}$ *for some* $B_1 \ge 2e^2 R$, $B_0 \ge 3$. *Then for all* $t \in [T - h]$,

$$|\left(\operatorname{D}_h (\phi \circ L)\right)^{(t)}| \le \frac{12RQe^2}{B_1} \cdot \alpha^h \cdot h^{B_0 h + 1}.$$

*Proof of Lemma 4.5.* Note that the $h$th order finite differences of a constant sequence are identically 0 for $h \ge 1$, so by subtracting $\phi(\mathbf{0})$ from $\phi$, we may assume without loss of generality that $\phi(\mathbf{0}) = 0$. (Here **0** denotes the all-zeros vector.)

By assumption, the radius of convergence of the power series of $\phi$ at the origin is at least 1, and so for each $\gamma \in \mathbb{Z}_{\ge 0}^n$, there is a real number $a_\gamma$ so that for $z = (z_1, \ldots, z_n)$ with $|z_j| \le 1$ for each $j$,

$$\phi(z) = \sum_{k \in \mathbb{N}, \gamma \in \mathbb{Z}_{\ge 0}^n : |\gamma| = k} a_\gamma z^\gamma. \tag{50}$$

Let $A_k := \sum_{\gamma \in \mathbb{Z}_{\ge 0}^n : |\gamma| = k} |a_\gamma|$; by the assumption that $\phi$ is $(Q, R)$-bounded, we have that $A_k \le Q \cdot R^k$ for all $k \in \mathbb{N}$.

For $\gamma \in \mathbb{Z}_{\ge 0}^n$, recall that $L^\gamma$ denotes the sequence $((L^\gamma)^{(1)}, \ldots, (L^\gamma)^{(T)})$, defined by $(L^\gamma)^{(t)} = (L^{(t)}(1))^{\gamma_1} \cdots (L^{(t)}(n))^{\gamma_n}$. Then since $\|L^{(t)}\|_\infty \le 1$ for all $t \in [T]$, we have that, for $t \in [T - h]$, $\left(\operatorname{D}_h (\phi \circ L)\right)^{(t)} = \sum_{\gamma \in \mathbb{Z}_{\ge 0}^n} a_\gamma \cdot (\operatorname{D}_h L^\gamma)^{(t)}$.

We next upper bound the quantities $|(\operatorname{D}_h L^\gamma)^{(t)}|$. To do so, fix some $\gamma \in \mathbb{Z}_{\ge 0}^n$, and set $k = |\gamma|$. For each function $\pi : [h] \to [k]$ and $r \in [k]$, recall the integers $h'_{\pi,r} \in \{0, 1, \ldots, h\}$, $t'_{\pi,r} \ge 0$, $j'_{\pi,r} \in [n]$ defined in Lemma 4.6. By assumption it holds that for each $t \in [T - h]$, each $h' \le h$, each $0 \le t' \le h$, $|(\operatorname{D}_{h'} L(j))^{(t+t')}| \le \frac{1}{B_1} \cdot \alpha^{h'} \cdot (h')^{B_0 h'}$. It follows that for each $t \in [T - h]$ and function $\pi : [h] \to [k]$,

$$\left| \prod_{r=1}^k \left( \operatorname{E}_{t'_{\pi,r}} \operatorname{D}_{h'_{\pi,r}} L(j'_{\pi,r}) \right)^{(t)} \right| \le \prod_{r=1}^k \frac{1}{B_1} \cdot \alpha^{h'_{\pi,r}} \cdot (h'_{\pi,r})^{B_0 h'_{\pi,r}} = \frac{\alpha^h}{B_1^k} \cdot \prod_{r=1}^k (h'_{\pi,r})^{B_0 h'_{\pi,r}},$$

where the last equality uses that $\sum_{r=1}^k h'_{\pi,r} = h$ (item 1 of Lemma 4.6). Then by Lemma 4.6, we have:

$$\left| (\operatorname{D}_h L^\gamma)^{(t)} \right| \le \sum_{\pi : [h] \to [k]} \left| \prod_{r=1}^k \left( \operatorname{E}_{t'_{\pi,r}} \operatorname{D}_{h'_{\pi,r}} L(j'_{\pi,r}) \right)^{(t)} \right|$$

$$\le \frac{\alpha^h}{B_1^k} \sum_{\pi : [h] \to [k]} \prod_{r=1}^k (h'_{\pi,r})^{B_0 h'_{\pi,r}}$$

$$\le \frac{\alpha^h}{B_1^k} \cdot h^{B_0 h} \max \left\{ k^7, (hk + 1) \cdot \exp \left( \frac{2k}{h^{B_0 - 1}} \right) \right\}, \tag{51}$$

where (51) follows from Lemma B.4, the fact that $B_0 \ge 3$, and that $h'_{\pi,r} = |\{q \in [h] : \pi(q) = r\}|$ (item 1 of Lemma 4.6).

We may now bound the order-$h$ finite differences of the sequence $\phi \circ L$ as follows: for $t \in [T - h]$,

$$\left| (D_h (\phi \circ L))^{(t)} \right| \leq \sum_{\gamma \in \mathbb{Z}_{\geq 0}^n} |a_\gamma| \cdot \left| (D_h L^\gamma)^{(t)} \right|$$

$$\leq \alpha^h \cdot h^{B_0 h + 1} \sum_{\gamma \in \mathbb{Z}_{\geq 0}^n} |a_\gamma| \cdot B_1^{-|\gamma|} \cdot \max \left\{ |\gamma|^7, (|\gamma| + 1) \cdot \exp \left( \frac{2|\gamma|}{h^{B_0 - 1}} \right) \right\} \tag{52}$$

$$\leq \alpha^h \cdot h^{B_0 h + 1} \cdot \sum_{k \in \mathbb{N}} A_k \cdot B_1^{-k} \cdot \left( k^7 + 2k \cdot \exp(2k/h^{B_0 - 1}) \right)$$

$$\leq \alpha^h \cdot h^{B_0 h + 1} \cdot Q \cdot \left( \sum_{k \in \mathbb{N}} k^7 \cdot (R/B_1)^k + \sum_{k \in \mathbb{N}} 2k \cdot (R/B_1)^k \cdot e^{2k} \right) \tag{53}$$

$$\leq \frac{2RQe^2}{B_1} \cdot \alpha^h \cdot h^{B_0 h + 1} \cdot \left( \sum_{k \in \mathbb{N}} k^7 \cdot (2e^2)^{-k} + 2 \sum_{k \in \mathbb{N}} k \cdot 2^{-k} \right) \tag{54}$$

$$= \frac{12RQe^2}{B_1} \cdot \alpha^h \cdot h^{B_0 h + 1}.$$

where (52) uses (51), (53) uses the bound $A_k \leq QR^k$, and (54) uses the assumption $B_1 \geq 2e^2 R$. This gives the desired conclusion of the lemma. □

## B.4  Proof of Lemma 4.4

In this section we prove Lemma 4.4. The detailed version of Lemma 4.4 is stated below.

**Lemma 4.4** (Detailed). *Fix a parameter $\alpha \in \left( 0, \frac{1}{H+3} \right)$. If all players follow Optimistic Hedge updates with step size $\eta \leq \frac{\alpha}{36e^5 m}$, then for any player $i \in [m]$, integer $h$ satisfying $0 \leq h \leq H$, time step $t \in [T - h]$, it holds that*

$$\| (D_h \ell_i)^{(t)} \|_\infty \leq \alpha^h \cdot h^{3h+1}.$$

*Proof.* We have that for each agent $i \in [m]$, each $t \in [T]$, and each $a_i \in [n_i]$, $\ell_i^{(t)}(a_i) = \mathbb{E}_{a_{i'} \sim x_{i'}^{(t)}: i' \neq i}[\mathcal{L}_i(a_1, \ldots, a_m)]$. Thus, for $1 \leq t \leq T$,

$$\left| (D_h \ell_i)^{(t)}(a_i) \right| = \left| \sum_{s=0}^{h} \binom{h}{s} (-1)^{h-s} \ell_i^{(t+s)}(a_i) \right| \tag{55}$$

$$= \left| \sum_{a_{i'} \in [n_{i'}], \, \forall i' \neq i} \mathcal{L}_i(a_1, \ldots, a_m) \sum_{s=0}^{h} \binom{h}{s} (-1)^{h-s} \cdot \prod_{i' \neq i} x_{i'}^{(t+s)}(a_{i'}) \right|$$

$$\leq \sum_{a_{i'} \in [n_{i'}], \, \forall i' \neq i} \left| \sum_{s=0}^{h} \binom{h}{s} (-1)^{h-s} \cdot \prod_{i' \neq i} x_{i'}^{(t+s)}(a_{i'}) \right|$$

$$= \sum_{a_{i'} \in [n_{i'}], \, \forall i' \neq i} \left| \left( D_h \left( \prod_{i' \neq i} x_{i'}(a_{i'}) \right) \right)^{(t)} \right|, \tag{56}$$

where (55) and (56) use Remark 4.3 and in (56), $\prod_{i' \neq i} x_{i'}(a_{i'})$ refers to the sequence $\prod_{i' \neq i} x_{i'}^{(1)}(a_{i'})$, $\prod_{i' \neq i} x_{i'}^{(2)}(a_{i'}), \ldots, \prod_{i' \neq i} x_{i'}^{(T)}(a_{i'})$.

In the remainder of this lemma we will prepend to the loss sequence $\ell_i^{(1)}, \ldots, \ell_i^{(T)}$ the vectors $\ell_i^{(0)} = \ell_i^{(-1)} := \mathbf{0} \in \mathbb{R}^{n_i}$. We will also prepend $x_i^{(0)} := x_i^{(1)} = (1/n_i, \ldots, 1/n_i) \in \Delta^{n_i}$ to the strategy sequence $x_i^{(1)}, \ldots, x_i^{(T)}$. Next notice that for any agent $i \in [m]$, any $t_0 \in \{0, 1, \ldots, T\}$, and

any $t \geq 0$, by the definition (1) of the Optimistic Hedge updates, it holds that, for each $j \in [n_i]$,

$$x_i^{(t_0+t+1)}(j) = \frac{x_i^{(t_0)}(j) \cdot \exp\left(\eta \cdot \left(\ell_i^{(t_0-1)}(j) - \sum_{s=0}^{t} \ell_i^{(t_0+s)}(j) - \ell_i^{(t_0+t)}(j)\right)\right)}{\sum_{k=1}^{n_i} x_i^{(t_0)}(k) \cdot \exp\left(\eta \cdot \left(\ell_i^{(t_0-1)}(k) - \sum_{s=0}^{t} \ell_i^{(t_0+s)}(k) - \ell_i^{(t_0+t)}(k)\right)\right)}.$$

Note in particular that our definitions of $\ell_i^{(0)}, \ell_i^{(-1)}, x_i^{(0)}$ ensure that the above equation holds even for $t_0 \in \{0, 1\}$. Now an integer $t_0$ satisfying $0 \leq t_0 \leq T$; for $t \geq 0$, let us write

$$\bar{\ell}_{i,t_0}^{(t)} := \ell_i^{(t_0-1)} - \sum_{s=0}^{t-1} \ell_i^{(t_0+s)} - \ell_i^{(t_0+t-1)}.$$

Also, for a vector $z = (z(1), \ldots, z(n_i)) \in \mathbb{R}^{n_i}$ and an index $j \in [n_i]$, define

$$\phi_{t_0,j}(z) := \frac{\exp(z(j))}{\sum_{k=1}^{n_i} x_i^{(t_0)}(k) \cdot \exp(z(k))}, \tag{57}$$

so that $x_i^{(t_0+t)}(j) = x_i^{(t_0)}(j) \cdot \phi_{t_0,j}(\eta \cdot \bar{\ell}_{i,t_0}^{(t)})$ for $t \geq 1$. In particular, for any $i \in [m]$, and any choices of $a_{i'} \in [n_{i'}]$ for all $i' \neq i$,

$$\prod_{i' \neq i} x_{i'}^{(t_0+t)}(a_{i'}) = \prod_{i' \neq i} x_{i'}^{(t_0)}(a_{i'}) \cdot \phi_{t_0,a_{i'}}(\eta \cdot \bar{\ell}_{i',t_0}^{(t)}). \tag{58}$$

Next, note that

$$\left(\mathrm{D}_1 \bar{\ell}_{i,t_0}\right)^{(t)} = \ell_i^{(t_0+t-1)} - 2\ell_i^{(t_0+t)} = \ell_i^{(t_0+t-1)} - 2\left(\mathrm{E}_1 \ell_i\right)^{(t_0+t-1)},$$

meaning that for any $h' \geq 1$,

$$\left(\mathrm{D}_{h'} \bar{\ell}_{i,t_0}\right)^{(t)} = \left(\mathrm{D}_{h'-1} \ell_i\right)^{(t_0+t-1)} - 2\left(\mathrm{E}_1 \mathrm{D}_{h'-1} \ell_i\right)^{(t_0+t-1)}. \tag{59}$$

We next establish the following claims which will allow us to prove Lemma 4.4 by induction.

**Claim B.8.** *For any $t_0 \in \{0, 1, \ldots, T\}$, $t \geq 0$, and $i \in [m]$, it holds that $\|\bar{\ell}_{i,t_0}^{(t)}\|_\infty \leq t + 2$.*

*Proof of Claim B.8.* The claim is immediate from the triangle inequality and the fact that $\|\ell_i^{(t)}\|_\infty \leq 1$ for all $t \in [T]$. $\square$

**Claim B.9.** *Fix $h$ so that $1 \leq h \leq H$. Suppose that for some $B_0 \geq 3$ and for all $0 \leq h' < h$, all $i \in [m]$, and all $t \leq T - h'$, it holds that $\|(\mathrm{D}_{h'} \ell_i)^{(t)}\|_\infty \leq \alpha^{h'} \cdot (h'+1)^{B_0(h'+1)}$. Suppose that the step size $\eta$ satisfies $\eta \leq \min\left\{\frac{\alpha}{36e^5 m}, \frac{1}{12e^5(H+3)m}\right\}$. Then for all $i \in [m]$ and $1 \leq t \leq T - h$,*

$$\left\|(\mathrm{D}_h \ell_i)^{(t)}\right\|_\infty \leq \alpha^h \cdot h^{B_0 h+1}. \tag{60}$$

*Proof of Claim B.9.* Set $B_1 := 12e^5 m$, so that the assumption of the claim gives $\eta \leq \min\left\{\frac{\alpha}{3B_1}, \frac{1}{B_1(H+3)}\right\}$.

We first use Lemma 4.5 to bound, for each $0 \leq t_0 \leq T - h$, $i \in [m]$, and $a_{i'} \in [n_{i'}]$ for all $i' \neq i$, the quantity $\left|\left(\mathrm{D}_h\left(\prod_{i' \neq i} x_{i'}(a_{i'})\right)\right)^{(t_0+1)}\right|$. In particular, we will apply Lemma 4.5 with $n = \sum_{i' \neq i} n_{i'}$, the value of $h$ in the statement of Claim B.9, $T = h + 1$, and the sequence $L^{(t)}$, for $1 \leq t \leq h + 1$, defined as

$$L^{(t)} = \left(\eta \cdot \bar{\ell}_{1,t_0}^{(t)}, \ldots, \eta \cdot \bar{\ell}_{i-1,t_0}^{(t)}, \eta \cdot \bar{\ell}_{i+1,t_0}^{(t)}, \ldots, \eta \cdot \bar{\ell}_{m,t_0}^{(t)}\right),$$

namely the concatenation of the vectors $\eta \cdot \bar{\ell}_{1,t_0}^{(t)}, \ldots, \eta \cdot \bar{\ell}_{i-1,t_0}^{(t)}, \eta \cdot \bar{\ell}_{i+1,t_0}^{(t)}, \ldots, \eta \cdot \bar{\ell}_{m,t_0}^{(t)}$. The function $\phi$ in Lemma 4.5 is set to the function that takes as input the concatenation of $z_{i'} \in \mathbb{R}^{n_{i'}}$ for all $i' \neq i$ and outputs:

$$\phi_{t_0, a_{-i}}(z_1, \ldots, z_{i-1}, z_{i+1}, \ldots, z_m) := \prod_{i' \neq i} \phi_{t_0, a_{i'}}(z_{i'}), \tag{61}$$

where the function $\phi_{t_0, a_{i'}}$ are as defined in (57). We first verify the preconditions of Lemma 4.5. By Lemma B.6, $\phi_{t_0, a_{-i}}$ is a $(1, e^3 m)$-bounded function for some constant $C \geq 1$. By Lemma B.7, the radius of convergence of each function $\phi_{t_0, a_{i'}}$ at $\mathbf{0}$ is at least 1; thus the radius of convergence of $\phi_{t_0, a_{-i}}$ at $\mathbf{0}$ is at least 1. Claim B.8 gives that $\|\bar{\ell}_{i,t_0}^{(t)}\|_\infty \leq t + 2 \leq h + 3$ for all $t \leq h + 1$. Thus, since $\eta \leq \frac{1}{B_1(H+3)}$,

$$\left\| \left( \mathrm{D}_0 \left( \eta \cdot \bar{\ell}_{i,t_0} \right) \right)^{(t)} \right\|_\infty = \| \eta \cdot \bar{\ell}_{i,t_0}^{(t)} \|_\infty \leq \eta \cdot (H+3) \leq \frac{1}{B_1}$$

for $1 \leq t \leq h_0 + 1$. Next, for $1 \leq h' \leq h$ and $1 \leq t \leq h + 1 - h'$, we have

$$\left\| \left( \mathrm{D}_{h'} \left( \eta \cdot \bar{\ell}_{i,t_0} \right) \right)^{(t)} \right\|_\infty \leq \eta \cdot \left\| \left( \mathrm{D}_{h'-1} \ell_i \right)^{(t_0+t-1)} \right\|_\infty + 2\eta \cdot \left\| \left( \mathrm{D}_{h'-1} \ell_i \right)^{(t_0+t)} \right\|_\infty \tag{62}$$

$$\leq 3\eta \cdot \alpha^{h'-1} \cdot (h')^{B_0(h')} \tag{63}$$

$$\leq \frac{1}{B_1} \cdot \alpha^{h'} \cdot (h')^{B_0(h')}, \tag{64}$$

where (62) follows from (59), (63) follows from the assumption in the statement of Claim B.9 and $t_0 + t + h' - 1 \leq t_0 + h \leq T$, and (64) follows from the fact that $3\eta \leq \frac{\alpha}{B_1}$. It then follows from Lemma 4.5 and (58) that

$$\frac{1}{\prod_{i' \neq i} x_{i'}^{(t_0)}(a_{i'})} \cdot \left| \left( \mathrm{D}_h \left( \prod_{i' \neq i} x_{i'}(a_{i'}) \right) \right)^{(t_0+1)} \right|$$

$$= \left| \left( \mathrm{D}_h \left( \prod_{i' \neq i} \left( \phi_{t_0, a_{i'}} \circ \eta \bar{\ell}_{i', t_0} \right) \right) \right)^{(t_0+1)} \right| \tag{65}$$

$$= \left| \left( \mathrm{D}_h \left( \phi_{t_0, a_{-i}} \circ \left( \eta \bar{\ell}_{1,t_0}, \ldots, \eta \bar{\ell}_{i-1,t_0}, \eta \bar{\ell}_{i+1,t_0}, \ldots, \eta \bar{\ell}_{m,t_0} \right) \right) \right)^{(1)} \right| \tag{66}$$

$$\leq \frac{12 e^5 m}{B_1} \cdot \alpha^h \cdot h^{B_0 h + 1} = \alpha^h \cdot (h)^{B_0 h + 1}. \tag{67}$$

(In particular, (65) uses (58), (66) uses the definition of $\phi_{t_0, a_{-i}}$ in (61), and (67) uses Lemma 4.5.)

Next we use (56), which gives that for each $i \in [m]$ and $t \geq 1$,

$$\left\| \left( \mathrm{D}_h \ell_i \right)^{(t)} \right\|_\infty \leq \sum_{a_{i'} \in [n_{i'}], \, \forall i' \neq i} \left| \left( \mathrm{D}_h \left( \prod_{i' \neq i} x_{i'}(a_{i'}) \right) \right)^{(t)} \right|$$

$$\leq \sum_{a_{i'} \in [n_{i'}], \, \forall i' \neq i} \prod_{i' \neq i} x_{i'}^{(t_0)}(a_{i'}) \cdot \alpha^h \cdot (h)^{B_0 h + 1} \tag{68}$$

$$= \alpha^h \cdot (h)^{B_0 h + 1},$$

where (68) follows from (67) with $t = t_0 + 1$ (here we use that $t_0$ may be 0). This completes the proof of Claim B.9. $\qquad\square$

It is immediate that for all $i \in [m], t \in [T]$, we have that $\| \left( \mathrm{D}_0 \ell_i \right)^{(t)} \|_\infty \leq 1 = \alpha^0 \cdot 1^{B_0 \cdot 1}$. We now apply Claim B.9 inductively with $B_0 = 3$, for which it suffices to have $\eta \leq \frac{\alpha}{36 e^5 m}$ as long as $\alpha \leq 1/(H+3)$. This gives that for $0 \leq h \leq H$, $i \in [m]$, and $t \in [T - h]$, $\| \left( \mathrm{D}_h \ell_i \right)^{(t)} \|_\infty \leq \alpha^h \cdot h^{3h+1}$, completing the proof of Lemma 4.4. $\qquad\square$

# C Proofs for Section 4.4

The main goal of this section is to prove Lemma 4.7. First, in Section C.1 we prove some preliminary lemmas and then we prove Lemma 4.7 in Section C.2

## C.1 Preliminary lemmas

Lemma C.1 shows that $\mathrm{Var}_P(W)$ and $\mathrm{Var}_{P'}(W)$ are close when the entries of $P, P'$ are close; it will be applied with $P, P'$ equal to the strategies $x_i^{(t)} \in \Delta^{n_i}$ played in the course of Optimistic Hedge.

**Lemma C.1.** *Suppose $n \in \mathbb{N}$ and $M > 0$ are given, and $W \in \mathbb{R}^n$ is a vector. Suppose $P, P' \in \Delta^n$ are distributions with* $\max\left\{\left\|\frac{P}{P'}\right\|_\infty, \left\|\frac{P'}{P}\right\|_\infty\right\} \leq 1 + \alpha$ *for some $\alpha > 0$. Then*

$$(1 - \alpha)\,\mathrm{Var}_P(W) \leq \mathrm{Var}_{P'}(W) \leq (1 + \alpha)\,\mathrm{Var}_P(W). \tag{69}$$

*Proof.* We first prove that $\mathrm{Var}_{P'}(W) \leq (1 + \alpha)\,\mathrm{Var}_P(W)$. To do so, note that since adding a constant to every entry of $W$ does not change $\mathrm{Var}_P(W)$ or $\mathrm{Var}_{P'}(W)$, by replacing $W$ with $W - \langle P, W \rangle \cdot \mathbf{1}$, we may assume without loss of generality that $\langle P, W \rangle = 0$. Thus $\mathrm{Var}_P(W) = \sum_{j=1}^n P(j)W(j)^2$. Now we may compute:

$$\begin{aligned}
\mathrm{Var}_{P'}(W) &\leq \sum_j P'(j) \cdot W(j)^2 \\
&= \sum_j P(j) \cdot W(j)^2 + \sum_j (P'(j) - P(j)) \cdot W(j)^2 \\
&= (1 + \alpha)\,\mathrm{Var}_P(W),
\end{aligned} \tag{70}$$

where (70) uses the fact that $\left\|\frac{P'}{P}\right\|_\infty \leq 1 + \alpha$.

By interchanging the roles of $P, P'$, we obtain that

$$\mathrm{Var}_{P'}(W) \geq \frac{1}{1 + \alpha}\,\mathrm{Var}_P(W) \geq (1 - \alpha)\,\mathrm{Var}_P(W).$$

This completes the proof of the lemma. $\qquad\square$

Next we prove Lemma 4.8 (recall that only the special case $\mu = 0$ was proved in Section 4.4). For convenience the lemma is repeated below.

**Lemma 4.8** (Restated). *Suppose $\mu \in \mathbb{R}$, $\alpha > 0$, and $W^{(0)}, \ldots, W^{(S-1)} \in \mathbb{R}$ is a sequence of reals satisfying*

$$\sum_{t=0}^{S-1} \left((\mathrm{D}_2^\circ W)^{(t)}\right)^2 \leq \alpha \cdot \sum_{t=0}^{S-1} \left((\mathrm{D}_1^\circ W)^{(t)}\right)^2 + \mu. \tag{71}$$

*Then*

$$\sum_{t=0}^{S-1} \left((\mathrm{D}_1^\circ W)^{(t)}\right)^2 \leq \alpha \cdot \sum_{t=1}^{S-1} (W^{(t)})^2 + \mu/\alpha.$$

To prove Lemma 4.8 we need the following basic facts about the Fourier transform:

**Fact C.2** (Parseval's equality). *It holds that $\sum_{t=0}^{S-1} |W^{(t)}|^2 = \frac{1}{S} \sum_{s=0}^{S-1} |\widehat{W}^{(s)}|^2$.*

The second fact gives a formula for the Fourier transform of the circular finite differences; its simple form is the reason we work with *circular* finite differences in this section:

**Fact C.3.** *For $h \in \mathbb{Z}_{\geq 0}$, $\widehat{\mathrm{D}_h^\circ W}^{(s)} = \widehat{W}^{(s)} \cdot (e^{2\pi i s t/S} - 1)^h$.*

*Proof of Lemma 4.8.* Note that the discrete Fourier transform of $D_1^\circ W$ satisfies $\widehat{D_1^\circ W}^{(s)} = \widehat{W}^{(s)} \cdot$
$(e^{2\pi i s/T} - 1)$, and similarly $\widehat{D_2^\circ W}^{(s)} = \widehat{W}^{(s)} \cdot (e^{2\pi i s/T} - 1)^2$, for $0 \le s \le S-1$. By the Cauchy-Schwarz inequality, Parseval's equality (Fact C.2), Fact C.3, and the assumption that (71) holds, we have

$$
\begin{aligned}
\sum_{t=0}^{S-1} \left( (D_1^\circ W)^{(t)} \right)^2 &= \frac{1}{S} \sum_{s=0}^{S-1} \left| \widehat{D_1^\circ W}^{(s)} \right|^2 \\
&= \frac{1}{S} \sum_{s=0}^{S-1} \left| \widehat{W}^{(s)} \cdot (e^{2\pi i s/T-1}) \right|^2 \\
&= \frac{1}{S} \sum_{s=0}^{S-1} \left| \widehat{W}^{(s)} \right| \cdot \left| \widehat{W}^{(s)} \right| \left| e^{2\pi i s/T} - 1 \right|^2 \\
&\le \sqrt{\frac{1}{S} \sum_{s=0}^{S-1} \left| \widehat{W}^{(s)} \right|^2} \cdot \sqrt{\frac{1}{S} \sum_{s=0}^{S-1} \left| \widehat{W}^{(s)} \right|^2 \cdot \left| e^{2\pi i s/T} - 1 \right|^4} \\
&= \sqrt{\sum_{t=0}^{S-1} (W^{(t)})^2} \cdot \sqrt{\frac{1}{S} \sum_{s=0}^{S-1} \left| \widehat{D_2^\circ W}^{(s)} \right|^2} \\
&= \sqrt{\sum_{t=0}^{S-1} (W^{(t)})^2} \cdot \sqrt{\sum_{t=0}^{S-1} \left( (D_2^\circ W)^{(t)} \right)^2} \\
&\le \sqrt{\sum_{t=0}^{S-1} (W^{(t)})^2} \cdot \sqrt{\alpha \cdot \sum_{t=0}^{S-1} \left( (D_1^\circ W)^{(t)} \right)^2 + \mu}. \quad (72)
\end{aligned}
$$

Note that for real numbers $A > 0$ and $\epsilon$ with $A + \epsilon > 0$, it holds that

$$
\frac{A^2}{A+\epsilon} = \frac{A}{1+\epsilon/A} \ge A \cdot (1 - \epsilon/A) = A - \epsilon.
$$

Taking $A = \sum_{t=0}^{S-1} \left( (D_1^\circ W)^{(t)} \right)^2$ and $\epsilon = \mu/\alpha$ (for which $A + \epsilon > 0$ is immediate) and using (72) then gives

$$
\sum_{t=0}^{S-1} \left( (D_1^\circ W)^{(t)} \right)^2 - \mu/\alpha \le \frac{\left( \sum_{t=0}^{S-1} \left( (D_1^\circ W)^{(t)} \right)^2 \right)^2}{\sum_{t=0}^{S-1} \left( (D_1^\circ W)^{(t)} \right)^2 + \mu/\alpha} \le \alpha \cdot \sum_{t=0}^{S-1} \left( W^{(t)} \right)^2,
$$

as desired. $\qquad\square$

## C.2  Proof of Lemma 4.7

Now we prove Lemma 4.7. For convenience we restate the lemma below with the exact value of the constant $C_0$ referred to in the version in Section 4.4.

**Lemma 4.7** (Restated). *For any $M, \zeta, \alpha > 0$ and $n \in \mathbb{N}$, suppose that $P^{(1)}, \dots, P^{(T)} \in \Delta^n$ and $Z^{(1)}, \dots, Z^{(T)} \in [-M, M]^n$ satisfy the following conditions:*

1. *The sequence $P^{(1)}, \dots, P^{(T)}$ is $\zeta$-consecutively close for some $\zeta \in [1/(2T), \alpha^4/8256]$.*

2. *It holds that $\sum_{t=1}^{T-2} \mathrm{Var}_{P^{(t)}} \left( (D_2 Z)^{(t)} \right) \le \alpha \cdot \sum_{t=1}^{T-1} \mathrm{Var}_{P^{(t)}} \left( (D_1 Z)^{(t)} \right) + \mu.$*

*Then*

$$
\sum_{t=1}^{T-1} \mathrm{Var}_{P^{(t)}} \left( (D_1 Z)^{(t)} \right) \le \alpha \cdot (1+\alpha) \sum_{t=1}^{T} \mathrm{Var}_{P^{(t)}} \left( Z^{(t)} \right) + \frac{\mu}{\alpha} + \frac{1290 M^2}{\alpha^3}. \quad (73)
$$

*Proof.* Fix a positive integer $S < 1/(2\zeta) < T$, to be specified exactly below. For $1 \leq t_0 \leq T - S + 1$, define $\mu_{t_0} \in \mathbb{R}$ by

$$\mu_{t_0} = \sum_{s=0}^{S-3} \mathrm{Var}_{P^{(t_0+s)}} \left( (\mathrm{D}_2 \, Z)^{(t_0+s)} \right) - \alpha \cdot \sum_{s=0}^{S-3} \mathrm{Var}_{P^{(t_0+s)}} \left( (\mathrm{D}_1 \, Z)^{(t_0+s)} \right). \tag{74}$$

Then

$$\sum_{t_0=1}^{T-S+1} \mu_{t_0}$$

$$= \sum_{t=1}^{T-2} \mathrm{Var}_{P^{(t)}} \left( (\mathrm{D}_2 \, Z)^{(t)} \right) \cdot \min\{S-2, t, T-t-1\}$$

$$- \alpha \cdot \sum_{t=1}^{T-1} \mathrm{Var}_{P^{(t)}} \left( (\mathrm{D}_1 \, Z)^{(t)} \right) \cdot \min\{S-2, t, T-t-1\}$$

$$\leq (S-2) \cdot \sum_{t=1}^{T-2} \mathrm{Var}_{P^{(t)}} \left( (\mathrm{D}_2 \, Z)^{(t)} \right) - (S-2)\alpha \cdot \sum_{t=1}^{T-1} \mathrm{Var}_{P^{(t)}} \left( (\mathrm{D}_1 \, Z)^{(t)} \right) + 8\alpha(S-2)^2 M^2$$

$$\tag{75}$$

$$\leq (S-2)\mu + 2\alpha(S-2)^2 M^2, \tag{76}$$

where (75) uses the fact that $\|Z^{(t)}\|_\infty \leq M$ and so $\|(\mathrm{D}_1 \, Z)^{(t)}\|_\infty \leq 2M$ for all $t \in [T]$, and the final inequality (76) follows from assumption 2 of the lemma statement.

By (74) and Lemma C.1 with $P = P^{(t_0)}$, we have, for some constant $C > 0$,

$$\sum_{s=0}^{S-3} \mathrm{Var}_{P^{(t_0)}} \left( (\mathrm{D}_2 \, Z)^{(t_0+s)} \right) \leq (1 + 2\zeta S) \cdot \sum_{s=0}^{S-3} \mathrm{Var}_{P^{(t_0+s)}} \left( (\mathrm{D}_2 \, Z)^{(t_0+s)} \right)$$

$$= (1 + 2\zeta S)\alpha \cdot \sum_{s=0}^{S-3} \mathrm{Var}_{P^{(t_0+s)}} \left( (\mathrm{D}_1 \, Z)^{(t_0+s)} \right) + (1 + 2\zeta S)\mu_{t_0}$$

$$\leq (1 + 2\zeta S)^2 \alpha \cdot \sum_{s=0}^{S-3} \mathrm{Var}_{P^{(t_0)}} \left( (\mathrm{D}_1 \, Z)^{(t_0+s)} \right) + (1 + 2\zeta S)\mu_{t_0}. \tag{77}$$

Here we have used that for $0 \leq s \leq S$, it holds that $\max \left\{ \left\| \frac{P^{(t_0+s)}}{P^{(t_0)}} \right\|_\infty, \left\| \frac{P^{(t_0)}}{P^{(t_0+s)}} \right\|_\infty \right\} \leq (1+\zeta)^S \leq 1 + 2\zeta S$ since $\zeta S \leq 1/2$.

For any integer $1 \leq t_0 \leq T - S + 1$, we define the sequence $Z_{t_0}^{(s)} := Z^{(t_0+s)} - \langle P^{(t_0)}, Z^{(t_0+s)} \rangle \mathbf{1}$, for $0 \leq s \leq S - 1$. Thus $\langle Z_{t_0}^{(s)}, P^{(t_0)} \rangle = 0$ for $0 \leq s \leq S - 1$, which implies that for all $h \geq 0$, $0 \leq s \leq S - 1$, $\langle (\mathrm{D}_h^\circ \, Z_{t_0})^{(s)}, P^{(t_0)} \rangle = 0$, and thus

$$\mathrm{Var}_{P^{(t_0)}} \left( (\mathrm{D}_h^\circ \, Z_{t_0})^{(s)} \right) = \sum_{j=1}^n P^{(t_0)}(j) \cdot (\mathrm{D}_h^\circ \, Z_{t_0})^{(s)}(j)^2. \tag{78}$$

By the definition of the sequence $Z_{t_0}$, for $0 \leq s \leq S - h - 1$, we have

$$\mathrm{Var}_{P^{(t_0)}} \left( (\mathrm{D}_h \, Z)^{(t_0+s)} \right) = \mathrm{Var}_{P^{(t_0)}} \left( (\mathrm{D}_h \, Z_{t_0})^{(s)} \right) = \mathrm{Var}_{P^{(t_0)}} \left( (\mathrm{D}_h^\circ \, Z_{t_0})^{(s)} \right). \tag{79}$$

For $1 \leq t_0 \leq T - S + 1$, let us now define

$$\nu_{t_0,j} := \sum_{s=0}^{S-1} (\mathrm{D}_2^\circ \, Z_{t_0})^{(s)}(j)^2 - (1 + 2\zeta S)^2 \alpha \cdot \sum_{s=0}^{S-1} (\mathrm{D}_1^\circ \, Z_{t_0})^{(s)}(j)^2, \tag{80}$$

so that, by (77), (78), and (79),

$$\sum_{j=1}^{n} P^{(t_0)}(j) \cdot \nu_{t_0,j}$$

$$= \sum_{s=0}^{S-1} \mathrm{Var}_{P^{(t_0)}} \left( (\mathrm{D}_2^{\circ} Z_{t_0})^{(s)} \right) - (1 + 2\zeta S)^2 \alpha \cdot \sum_{s=0}^{S-1} \mathrm{Var}_{P^{(t_0)}} \left( (\mathrm{D}_1^{\circ} Z_{t_0})^{(s)} \right)$$

$$\leq \left( \sum_{s=0}^{S-3} \mathrm{Var}_{P^{(t_0)}} \left( (\mathrm{D}_2 Z)^{(t_0+s)} \right) \right) + \mathrm{Var}_{P^{(t_0+1)}} \left( (\mathrm{D}_2^{\circ} Z)^{(t_0+S-2)} \right) + \mathrm{Var}_{P^{(t_0+1)}} \left( (\mathrm{D}_2^{\circ} Z)^{(t_0+S-1)} \right)$$

$$- (1 + 2\zeta S)^2 \alpha \cdot \sum_{s=0}^{S-3} \mathrm{Var}_{P^{(t_0)}} \left( (\mathrm{D}_1 Z)^{(t_0+s)} \right)$$

$$\leq (1 + 2\zeta S)\mu_{t_0} + \mathrm{Var}_{P^{(t_0)}} \left( (\mathrm{D}_2^{\circ} Z)^{(t_0+S-2)} \right) + \mathrm{Var}_{P^{(t_0)}} \left( (\mathrm{D}_2^{\circ} Z)^{(t_0+S-1)} \right). \tag{81}$$

By (80) and Lemma 4.8 applied to the sequence $Z_{t_0}^{(0)}, \ldots, Z_{t_0}^{(S-1)}$, it holds that, for each $j \in [n]$,

$$\sum_{s=0}^{S-1} (\mathrm{D}_1^{\circ} Z_{t_0})^{(s)} (j)^2 \leq (1 + \zeta C S)^2 \alpha \cdot \sum_{s=0}^{S-1} Z_{t_0}^{(s)}(j)^2 + \frac{\nu_{t_0,j}}{(1 + \zeta C S)^2 \alpha}. \tag{82}$$

Then we have:

$$\sum_{s=0}^{S-2} \mathrm{Var}_{P^{(t_0)}} \left( (\mathrm{D}_1 Z)^{(t_0+s)} \right)$$

$$= \sum_{s=0}^{S-2} \mathrm{Var}_{P^{(t_0)}} \left( (\mathrm{D}_1^{\circ} Z_{t_0})^{(s)} \right) \tag{83}$$

$$\leq (1 + 2\zeta S)^2 \alpha \cdot \sum_{s=0}^{S-1} \mathrm{Var}_{P^{(t_0)}} \left( Z_{t_0}^{(s)} \right) + \sum_{j=1}^{n} P^{(t_0)}(j) \cdot \frac{\nu_{t_0,j}}{(1 + 2\zeta S)^2 \alpha} \tag{84}$$

$$\leq (1 + 2\zeta S)^2 \alpha \sum_{s=0}^{S-1} \mathrm{Var}_{P^{(t_0)}} \left( Z_{t_0}^{(s)} \right) + \frac{\mu_{t_0}}{(1 + 2\zeta S)\alpha} + \frac{\mathrm{Var}_{P^{(t_0)}} \left( (\mathrm{D}_2^{\circ} Z)^{(t_0+S-2)} \right) + \mathrm{Var}_{P^{(t_0)}} \left( (\mathrm{D}_2^{\circ} Z)^{(t_0+S-1)} \right)}{(1 + 2\zeta S)^2 \alpha},$$

$$\tag{85}$$

where (83) follows from (79), (84) follws from (82) and (78), and (85) follows from (81). Summing the above for $1 \le t_0 \le T - S + 1$, we obtain, for some constant $C > 0$,

$$(S - 1) \cdot \sum_{t=1}^{T-1} \mathrm{Var}_{P^{(t)}} \left( (\mathrm{D}_1 Z)^{(t)} \right)$$

$$\le \sum_{t_0=1}^{T-S+1} \sum_{s=0}^{S-2} \mathrm{Var}_{P^{(t_0+s)}} \left( (\mathrm{D}_1 Z)^{(t_0+s)} \right) + 8(S-1)^2 M^2 \tag{86}$$

$$\le \sum_{t_0=1}^{T-S+1} (1 + 2\zeta S) \sum_{s=0}^{S-2} \mathrm{Var}_{P^{(t_0)}} \left( (\mathrm{D}_1 Z)^{(t_0+s)} \right) + 8(S-1)^2 M^2 \tag{87}$$

$$\le (1 + 2\zeta S)^3 \alpha \sum_{t_0=1}^{T-S+1} \sum_{s=0}^{S-1} \mathrm{Var}_{P^{(t_0)}} \left( Z_{t_0}^{(s)} \right) + \sum_{t_0=1}^{T-S+1} \frac{\mu_{t_0}}{\alpha} + 8(S-1)^2 M^2$$

$$+ \sum_{t_0=1}^{T-S+1} \frac{\mathrm{Var}_{P^{(t_0)}} \left( (\mathrm{D}_2^{\circ} Z)^{(t_0+S-2)} \right) + \mathrm{Var}_{P^{(t_0)}} \left( (\mathrm{D}_2^{\circ} Z)^{(t_0+S-1)} \right)}{(1 + 2\zeta S)\alpha} \tag{88}$$

$$\le (1 + 2\zeta S)^4 \alpha \sum_{t_0=1}^{T-S+1} \sum_{s=0}^{S-1} \mathrm{Var}_{P^{(t_0+s)}} \left( Z^{(t_0+s)} \right) + \sum_{t_0=1}^{T-S+1} \frac{\mu_{t_0}}{\alpha} + 8(S-1)^2 M^2$$

$$+ \frac{4}{(1 + 2\zeta S)\alpha} \sum_{t_0=1}^{T-S+1} \left[ \mathrm{Var}_{P^{(t_0)}} \left( Z^{(t_0+S-2)} \right) + 3 \mathrm{Var}_{P^{(t_0)}} \left( Z^{(t_0+S-1)} \right) \right.$$

$$\left. + 3 \mathrm{Var}_{P^{(t_0)}} \left( Z^{(t_0)} \right) + \mathrm{Var}_{P^{(t_0)}} \left( Z^{(t_0+1)} \right) \right] \tag{89}$$

$$\le (1 + 2\zeta S)^4 \alpha S \sum_{t=1}^{T} \mathrm{Var}_{P^{(t)}} \left( Z^{(t)} \right) + \sum_{t_0=1}^{T-S+1} \frac{\mu_{t_0}}{\alpha} + 8(S-1)^2 M^2$$

$$+ \frac{32}{(1 + 2\zeta S)\alpha} \sum_{t=1}^{T} \mathrm{Var}_{P^{(t)}} \left( Z^{(t)} \right) \tag{90}$$

$$\le (1 + 2\zeta S)^4 \alpha S \cdot \left( 1 + \frac{32}{\alpha^2 S} \right) \sum_{t=1}^{T} \mathrm{Var}_{P^{(t)}} \left( Z^{(t)} \right) + \sum_{t_0=1}^{T-S+1} \frac{\mu_{t_0}}{\alpha} + 8(S-1)^2 M^2 \tag{91}$$

$$\le (1 + 2\zeta S)^4 \alpha S \cdot \left( 1 + \frac{32}{\alpha^2 S} \right) \sum_{t=1}^{T} \mathrm{Var}_{P^{(t)}} \left( Z^{(t)} \right) + \frac{(S-2)\mu}{\alpha} + 10(S-1)^2 M^2, \tag{92}$$

where:

- (86) follows since $\|Z^{(t)}\|_\infty \le M$ and thus $\| (\mathrm{D}_1 Z)^{(t)} \|_\infty \le 2M$ for all $t$;

- (87) follows from Lemma C.1 and the fact that for $0 \le s \le S - 2$, $\max \left\{ \left\| \frac{P^{(t_0+s)}}{P^{(t_0)}} \right\|_\infty, \left\| \frac{P^{(t_0)}}{P^{(t_0+s)}} \right\|_\infty \right\} \le 1 + 2\zeta S$ as established above as a consequence of the fact that the distributions $P^{(t)}$ are $\zeta$-consecutively close.

- (88) follows from (85);

- The first term in (89) is bounded using Lemma C.1 and the fact that the distributions $P^{(t)}$ are $\zeta$-consecutively close, and the the final term in (89) is bounded using the fact that for any vectors $Z_1, \ldots, Z_k \in \mathbb{R}^n$ and any $P \in \Delta^n$, we have $\mathrm{Var}_P (Z_1 + \cdots + Z_k) \le k \cdot (\mathrm{Var}_P (Z_1) + \cdots + \mathrm{Var}_P (Z_k))$;

- (90) and (91) by rearranging terms;

- (92) follows from (76).

Now choose $S = \left\lceil \frac{128}{\alpha^3} \right\rceil$, so that $\frac{32}{\alpha^2 S} \leq \frac{\alpha}{4}$. Therefore, as long as $2\zeta S \leq \frac{\alpha}{32}$, we have, since $\alpha \leq 1/2$, that

$$(1 + 2\zeta S)^4 \alpha \cdot \frac{S}{S-1} \cdot \left(1 + \frac{32}{\alpha^2 S}\right) \leq \alpha \cdot (1 + \alpha/4)^3 \leq \alpha \cdot (1 + \alpha).$$

Then it follows from (92) that

$$\sum_{t=1}^{T-1} \mathrm{Var}_{P^{(t)}} \left( (\mathrm{D}_1\, Z)^{(t)} \right) \leq \alpha(1 + \alpha) \cdot \sum_{t=1}^{T} \mathrm{Var}_{P^{(t)}} \left( Z^{(t)} \right) + \frac{\mu}{\alpha} + 10SM^2, \qquad (93)$$

Using that $S \leq \frac{129}{\alpha^3}$, the inequality $2\zeta S \leq \alpha/32$ can be satisfied by ensuring that $\zeta \leq \frac{\alpha^4}{8256} = \frac{\alpha^4}{129 \cdot 42} \leq \frac{\alpha}{64S}$. Note that our choice of $S$ ensures that $\zeta S \leq 1/2$, as was assumed earlier. Moreover, we have $10SM^2 \leq \frac{1290 M^2}{\alpha^3}$. Thus, (93) gives the desired result. $\qquad \square$

## C.3 Completing the proof of Theorem 3.1

Using the lemma developed in the previous sections we now can complete the proof of Theorem 3.1.

We begin by proving Lemma 4.2. The lemma is restated formally below.

**Lemma 4.2** (Detailed). *There are constants $C, C' > 1$ so that the following holds. Suppose a time horizon $T \geq 4$ is given, we set $H := \lceil \log T \rceil$, and all players play according to Optimistic Hedge with step size $\eta$ satisfying $1/T \leq \eta \leq \frac{1}{C \cdot mH^4}$. Then for any $i \in [m]$, the losses $\ell_i^{(1)}, \ldots, \ell_i^{(T)} \in [0, 1]^{n_i}$ for player $i$ satisfy:*

$$\sum_{t=1}^{T} \mathrm{Var}_{x_i^{(t)}} \left( \ell_i^{(t)} - \ell_i^{(t-1)} \right) \leq \frac{1}{2} \cdot \sum_{t=1}^{T} \mathrm{Var}_{x_i^{(t)}} \left( \ell_i^{(t-1)} \right) + C'H^5. \qquad (94)$$

*Proof.* Since $T \geq 4$, we have that $H \geq 2$. Set $C_0 = 1290$ (note that $C_0$ is the constant appearing in the inequality (73) of the statement of Lemma 4.7 in Section C.2), and $C_1 = 8256$ (note that $C_1$ is the constant appearing in item 1 of Lemma 4.7 in Section C.2). Set $\alpha = 1/(4H)$, and $\alpha_0 = \frac{\sqrt{\alpha/8}}{H^3} < \frac{1}{H+3}$. Finally note that our assumption on $\eta$ implies that, as long as the constant $C$ satisfies $C \geq 4^4 \cdot 7 \cdot C_1 = 14794752$,

$$\eta \leq \min\left\{ \frac{\alpha^4}{7C_1}, \frac{\alpha_0}{36e^5 m} \right\}. \qquad (95)$$

By Lemma 4.4 with the parameter $\alpha$ in the lemma set to $\alpha_0$, for each $i \in [m]$, $0 \leq h \leq H$ and $1 \leq t \leq T - h$, it holds that $\left\| (\mathrm{D}_h\, \ell_i)^{(t)} \right\|_\infty \leq H \cdot \left( \alpha_0 H^3 \right)^h$. Also set $\mu = H \cdot \left( \alpha_0 H^3 \right)^H$. Therefore, we have, for each $i \in [m]$,

$$\sum_{t=1}^{T-H} \mathrm{Var}_{x_i^{(t)}} \left( (\mathrm{D}_H\, \ell_i)^{(t)} \right) \leq \alpha \cdot \sum_{t=1}^{T-H+1} \mathrm{Var}_{x_i^{(t)}} \left( (\mathrm{D}_{H-1}\, \ell_i)^{(t)} \right) + \mu^2 T. \qquad (96)$$

We will now prove, via reverse induction on $h$, that for all $i \in [m]$ and $h$ satisfying $H - 1 \geq h \geq 0$,

$$\sum_{t=1}^{T-h-1} \mathrm{Var}_{x_i^{(t)}} \left( (\mathrm{D}_{h+1}\, \ell_i)^{(t)} \right) \leq \alpha \cdot (1+2\alpha)^{H-h-1} \cdot \sum_{t=1}^{T-h} \mathrm{Var}_{x_i^{(t)}} \left( (\mathrm{D}_h\, \ell_i)^{(t)} \right) + \frac{2C_0 \cdot H^2 \cdot \left( 2\alpha_0 H^3 \right)^{2h}}{\alpha^3}. \qquad (97)$$

The base case $h = H - 1$ is verified by (96) and the fact that $2^{2(H-1)} \geq 2^H \geq T$. Now suppose that (97) holds for some $h$ satisfying $H - 1 \geq h \geq 1$. We will now apply Lemma 4.7, with $P^{(t)} = x_i^{(t)}$ and $Z^{(t)} = (\mathrm{D}_{h-1}\, \ell_i)^{(t)}$ for $1 \leq t \leq T - h + 1$, as well as $M = H \cdot \left( 2\alpha_0 H^3 \right)^{h-1}$, $\zeta = 7\eta$, $\mu = \frac{2C_0 \cdot H^2 \cdot \left( 2\alpha_0 H^3 \right)^{2h}}{\alpha^3}$, and the parameter $\alpha$ of Lemma 4.7 set to $\alpha \cdot (1 + 2\alpha)^{H-h-1}$. We first verify that precondition 1 holds. By the definition of the Optimistic Hedge updates, the sequence $x_i^{(1)}, \ldots, x_i^{(T)}$ is $\exp(6\eta)$-consecutively close, and thus $(1 + 7\eta)$-consecutively close (since $\exp(6\eta) \leq 1 + 7\eta$ for $\eta$ satisfying (95)). Our choice of $\eta, \alpha$ ensures that $1/(2(T - h + 1)) \leq 7\eta \leq \alpha^4/C_1$. This verifies

that condition 1 of Lemma 4.7 holds. Note that condition 2 of the lemma holds by (97) and our choice of $\mu$. Therefore, by Lemma 4.7 and the fact that $1 + \alpha \cdot (1 + 2\alpha)^H \leq 1 + 2\alpha$ for our choice of $\alpha = 1/(4H)$, it follows that

$$
\begin{aligned}
\sum_{t=1}^{T-h} \mathrm{Var}_{x_i^{(t)}}\left(\left(\mathrm{D}_h\, \ell_i\right)^{(t)}\right) \leq & \alpha \cdot (1 + 2\alpha)^{H-h} \cdot \sum_{t=1}^{T-h+1} \mathrm{Var}_{x_i^{(t)}}\left(\left(\mathrm{D}_{h-1}\, \ell_i\right)^{(t)}\right) + \frac{2C_0 \cdot H^2 \cdot \left(2\alpha_0 H^3\right)^{2h}}{\alpha^4} \\
& + \frac{C_0 \cdot H^2 \cdot \left(2\alpha_0 H^3\right)^{2(h-1)}}{\alpha^3} \\
\leq & \alpha \cdot (1 + 2\alpha)^{H-h} \cdot \sum_{t=1}^{T-h+1} \mathrm{Var}_{x_i^{(t)}}\left(\left(\mathrm{D}_{h-1}\, \ell_i\right)^{(t)}\right) \\
& + \frac{C_0 \cdot H^2 \cdot (2\alpha_0 H^3)^{2(h-1)}}{\alpha^3} \cdot \left(1 + \frac{2(2\alpha_0 H^3)^2}{\alpha}\right) \\
\leq & \alpha \cdot (1 + 2\alpha)^{H-h} \cdot \sum_{t=1}^{T-h+1} \mathrm{Var}_{x_i^{(t)}}\left(\left(\mathrm{D}_{h-1}\, \ell_i\right)^{(t)}\right) + \frac{2C_0 \cdot H^2 \cdot (2\alpha_0 H^3)^{2(h-1)}}{\alpha^3},
\end{aligned}
$$

where the final equality follows since $\alpha_0$ is chosen so that $2(2\alpha_0 H^3)^2 = \alpha$. This completes the proof of the inductive step. Thus (97) holds for $h = 0$. Using again that the sequence $x_i^{(t)}$ is $(1 + 7\eta)$-exponentially close, we see that

$$
\begin{aligned}
& \sum_{t=1}^{T} \mathrm{Var}_{x_i^{(t)}}\left(\ell_i^{(t)} - \ell_i^{(t-1)}\right) \\
\leq & 1 + \sum_{t=2}^{T} \mathrm{Var}_{x_i^{(t)}}\left(\ell_i^{(t)} - \ell_i^{(t-1)}\right) \\
\leq & 1 + (1 + 7\eta) \sum_{t=1}^{T-1} \mathrm{Var}_{x_i^{(t)}}\left(\left(\mathrm{D}_1\, \ell_i\right)^{(t)}\right) & (98) \\
\leq & 1 + (1 + 7\eta) \cdot \left(\alpha(1 + 2\alpha)^{H-1} \sum_{t=1}^{T} \mathrm{Var}_{x_i^{(t)}}\left(\ell_i^{(t)}\right) + \frac{2C_0 H^2}{\alpha^3}\right) & (99) \\
\leq & 2 + (1 + 7\eta) \cdot \left(\alpha(1 + 2\alpha)^{H-1}(1 + 7\eta) \sum_{t=2}^{T} \mathrm{Var}_{x_i^{(t)}}\left(\ell_i^{(t-1)}\right) + \frac{2C_0 H^2}{\alpha^3}\right) & (100) \\
\leq & \alpha(1 + 2\alpha)^{H} \sum_{t=1}^{T} \mathrm{Var}_{x_i^{(t)}}\left(\ell_i^{(t-1)}\right) + \frac{2(1 + 7\eta)C_0 H^2}{\alpha^3} + 2 \\
\leq & 2\alpha \sum_{t=1}^{T} \mathrm{Var}_{x_i^{(t)}}\left(\ell_i^{(t-1)}\right) + \frac{2(1 + 7\eta)C_0 H^2}{\alpha^3} + 2, & (101)
\end{aligned}
$$

where (98) and (100) follow from Lemma C.1, and (99) uses (97) for $h = 0$. Now, (101) verifies the statement of the lemma. In summary, it suffices to take $C = 14794752$ and $C' = 2 + 2(1 + 7/8256)C_0 \cdot 4^3 = 165262$. $\qquad\square$

We are finally ready to prove Theorem 3.1. For convenience the theorem is restated below.

**Theorem 3.1** (Restated). *There are constants $C, C' > 1$ so that the following holds. Suppose a time horizon $T \in \mathbb{N}$ is given. Suppose all players play according to Optimistic Hedge with any positive step size $\eta \leq \frac{1}{C \cdot m \log^4 T}$. Then for any $i \in [m]$, the regret of player $i$ satisfies*

$$
\mathrm{Reg}_{i,T} \leq \frac{\log n_i}{\eta} + C' \cdot \log T. \tag{102}
$$

*In particular, if the players' step size is chosen as $\eta = \frac{1}{C \cdot m \log^4 T}$, then the regret of player $i$ satisfies*

$$
\mathrm{Reg}_{i,T} \leq O\left(m \cdot \log n_i \cdot \log^4 T\right). \tag{103}
$$

*Proof.* The conclusion of the theorem is immediate if $T < 4$, so we may assume from here on that $T \geq 4$. Moreover, the conclusion of (102) is immediate if $\eta \leq 1/T$ (as $\text{Reg}_{i,T} \leq T$ necessarily), so we may also assume that $\eta \geq 1/T$. Let $C''$ be the constant $C$ of Lemma 4.1, let $B$ be the constant called $C$ in Lemma 4.2 and $B'$ be the constant called $C'$ in Lemma 4.2. As long as the constant $C$ of Theorem 3.1 is chosen so that $C \geq B$ and $\eta \leq \frac{1}{C \cdot m \log^4 T}$ implies that $C'' \eta \leq 1/6$, we have the following:

$$\text{Reg}_{i,T} \leq \frac{\log n_i}{\eta} + \sum_{t=1}^{T} \left( \frac{\eta}{2} + C\eta^2 \right) \text{Var}_{x_i^{(t)}} \left( \ell_i^{(t)} - \ell_i^{(t-1)} \right) - \sum_{t=1}^{T} \frac{(1 - C\eta)\eta}{2} \text{Var}_{x_i^{(t)}} \left( \ell_i^{(t-1)} \right) \tag{104}$$

$$\leq \frac{\log n_i}{\eta} + \frac{2\eta}{3} \sum_{t=1}^{T} \text{Var}_{x_i^{(t)}} \left( \ell_i^{(t)} - \ell_i^{(t-1)} \right) - \frac{\eta}{3} \sum_{t=1}^{T} \text{Var}_{x_i^{(t)}} \left( \ell_i^{(t-1)} \right)$$

$$\leq \frac{\log n_i}{\eta} + \frac{2\eta}{3} \cdot \left( B' \cdot (2 \log T)^5 \right) \tag{105}$$

$$\leq \frac{\log n_i}{\eta} + 32 B' \cdot \log T, \tag{106}$$

where (104) follows from Lemma 4.1, (105) follows from Lemma 4.2, and (106) follows from the upper bound $\eta \leq \frac{1}{Cm \log^4 T}$. We have thus established (102). The upper bound (103) follows immediately. $\qquad\square$

## D    Adversarial regret bounds

In this section we discuss how Optimistic Hedge can be modified to achieve an algorithm that obtains the fast rates of Theorem 3.1 when played by all players, and which still obtains the optimal rate of $O(\sqrt{T})$ in the adversarial setting. Such guarantees are common in the literature [DDK11, RS13b, SALS15, KHSC18, HAM21]. The guarantees of this modification of Optimistic Hedge are stated in the following corollary (of Lemmas 4.1 and 4.2):

**Corollary D.1.** *There is an algorithm $\mathcal{A}$ which, if played by all $m$ players in a game, achieves the regret bound of $\text{Reg}_{i,T} \leq O(m \cdot \log n_i \cdot \log^4 T)$ for each player $i$; moreover, when player $i$ is faced with an* adversarial *sequence of losses, the algorithm $\mathcal{A}$'s regret bound is $\text{Reg}_{i,T} \leq O(m \log n_i \cdot \log^4 T + \sqrt{T \log n_i})$.*

*Proof.* Let $C$ be the constant called $C$ in Theorem 3.1 and $C'$ be the constant called $C'$ in Lemma 4.2. The algorithm $\mathcal{A}$ of Corollary D.1 is obtained as follows:

1. Initially run Optimistic Hedge, with the step-size $\eta = \frac{1}{Cm \log^4 T}$.

2. If, for some $T_0 \geq 4$, (94) first fails to hold at time $T_0$, i.e.,

$$\sum_{t=1}^{T_0} \text{Var}_{x_i^{(t)}} \left( \ell_i^{(t)} - \ell_i^{(t-1)} \right) > \frac{1}{2} \cdot \sum_{t=1}^{T_0} \text{Var}_{x_i^{(t)}} \left( \ell_i^{(t-1)} \right) + C' \lceil \log T \rceil^5, \tag{107}$$

then set $\eta' = \sqrt{\frac{\log n_i}{T}}$ and continue on running Optimistic Hedge with step size $\eta'$.

If there is no $T_0 \geq 4$ so that (107) holds (and by Lemma 4.2, this will be the case when $\mathcal{A}$ is played by all $m$ players in a game), then the proof of Theorem 3.1 shows that the regret of each player $i$ is bounded as $\text{Reg}_{i,T} \leq O(m \log n_i \cdot \log^4 T)$. Otherwise, since $T_0$ is defined as the smallest integer at least 4 so that (107) holds, we have

$$\sum_{t=1}^{T_0} \text{Var}_{x_i^{(t)}} \left( \ell_i^{(t)} - \ell_i^{(t-1)} \right) \leq \frac{1}{2} \cdot \sum_{t=1}^{T_0} \text{Var}_{x_i^{(t)}} \left( \ell_i^{(t-1)} \right) + C' \lceil \log T \rceil^5 + 4,$$

and thus, by Lemma 4.1, for any $x^\star \in \Delta^{n_i}$,

$$\sum_{t=1}^{T_0} \langle \ell_i^{(t)}, x_i^{(t)} - x^\star \rangle \leq \mathrm{Reg}_{i,T_0} \leq O(m \log n_i \cdot \log^4 T_0). \tag{108}$$

Further, by the choice of step size $\eta' = \sqrt{\frac{\log n_i}{T}}$ for time steps $t > T_0$, we have, for any $x^\star \in \Delta^{n_i}$,

$$\sum_{t=T_0+1}^{T} \langle \ell_i^{(t)}, x_i^{(t)} - x^\star \rangle \leq \frac{\log n_i}{\eta'} + \eta' \sum_{t=T_0+1}^{T} \|\ell_i^{(t)} - \ell_i^{(t-1)}\|_\infty^2 \tag{109}$$

$$\leq \frac{\log n_i}{\eta'} + \eta' T \leq O(\sqrt{T \log n_i}), \tag{110}$$

where (109) uses [SALS15, Proposition 7]. Adding (108) and (110) completes the proof of the corollary. $\square$