# OpenReview forum: "Near-Optimal No-Regret Learning in General Games"
_NeurIPS.cc/2021/Conference — NeurIPS 2021 Oral_

### Official Review · Reviewer_uZix · 2021-07-05

**Rating:** 8
**Confidence:** 3

**Summary:**

The paper shows that if all the players of a multi-player general-sum game use Optimistic Hedge with sufficiently small step-size, the regret of each player is bounded as $O(\log^4 T)$.

**Limitations And Societal Impact:**

Fine

**Main Review:**

As far as I am aware, this paper is the first to prove that an algorithm enjoys $\text{poly}(\log T)$ regret in all multi-player general-sum games. I believe this is an important contribution. Moreover, in my eyes, the proofs are highly technical and non-trivial. This start with the use of local norms which are common for the analysis of online learning algorithms but were less considered for the analysis of Optimistic Hedge, and is followed by several amazing ideas which goes from finite difference sequences to discrete Fourier transform. Unfortunately, I was unable to go through the entire appendix, but the overall schema seems correct.

I would also like to highlight that, with a single theorem proven and fives pages dedicated to the outline of the proof, this paper seems to be quite atypical for NeurIPS. Nonetheless, I think the authors have made an effort to sketch their proof in the main text. Although the use of big O notations makes the statements of the theorems and the proofs a little problematic, the readers can already get an idea about the structure of the proof before potentially looking into the appendix for a more rigorous and complete version.

**Suggestions for potential improvements of the presentation**

Given the particularity of the paper, I am not really sure about how to further improve it. Below are just some personal thoughts that the authors can take as reference.

1. I found that the paper could benefit from a wrap-up section or paragraph. In fact, both sections 4.3 and 4.4 are technical and devoted to the proof of lemma 4.2 and are thus not directly related to the main result of the work. For sake of completeness, it would be nice to link these back to Theorem 3.1 at the end. (Sure, this may not add any extra value to the work. It is just a matter of taste.)

2. It seems that many constants mentioned in the theorems are in fact absolute constants that come from numerical inequalities. Maybe it is worth making this point clear to the audience, or even directly providing the numerical values for which the inequalities hold.

3. Typos
    - line 256: right-hand side?
    - line 267 and the subsequent inequality: the authors seem to confound $A_1$ and $A_2$
    - I quickly went through the proof of Lemma 4.1 and feel that we should put $\eta$ instead of $\eta/2$ both after $+$ and $-$ at the r.h.s of the inequality.

**Time Spent Reviewing:**

6+

---

> ### Author Response · Authors · 2021-08-10
> **Author Response**
>
> Thank you for your helpful suggestions on improving the structure of the paper. We will take them into consideration for the final version; in particular, we will incorporate a wrap-up section at the end.
>
> Regarding constants: The constants in our results are indeed explicit constants.
>
> Typos:
>
> - Line 256: yes, this should be right-hand side
>
> - Line 267: yes, there is a typo and we swapped $A_1$ and $A_2$.
>
> - In the proof of Lemma 4.1 it should be $\eta/2$. Though we start off in Eq. (28) with a factor of $\eta$, a factor of $\eta/2$ is subtracted in the course of the proof. Please see comment 1. for Reviewer wnsJ for further details.

---

### Official Review · Reviewer_wnsJ · 2021-07-15

**Rating:** 8
**Confidence:** 2

**Summary:**

This paper analyzes the regret of Optimistic Hedge in multi-player general-sum games. The paper shows that the algorithm attains polylog($T$) regret for each player, which improves over known rates that scale polynomially with $T$.

**Limitations And Societal Impact:**

This work is purely theoretical and do not present foreseeable negative social impact.

**Main Review:**

The focus of this paper is to analyze Optimistic Hedge (which can be seen as an instance of Optimistic Mirror Descent). The main contribution of this paper is the technical improvement in the analysis of this existing algorithm. The key technique, as I understand it, is to control the variance of slowly changing vectors via bounding the variance of its high-order finite differences. While the improvement in the regret bound is certainly substantial, I do have some concerns over the correctness of the results.

1. In Eq. (28) the proof of Lemma 4.1, the second term is
$\sum_{t=1}^T \Vert x_i^{(t)}-\tilde x_i^{(t)}\Vert^*_{x_i^{(t)}}\cdot  \sqrt{Var_{x^t_i}(l_i^{(t)}-l_i^{(t-1)})}$
. The dual norm term is then controlled in Eq. (32) by $\sqrt{(1+C_1\eta)\eta^2}\cdot \sqrt{Var_{x^t_i}(l_i^{(t)}-l_i^{(t-1)})}$. Thus the whole term should be controlled by $(1+O(1))\eta \cdot Var_{x^t_i}(l_i^{(t)}-l_i^{(t-1)})$. Yet in Eq. (27), this term is shown as $(\frac{1}{2}+O(1))\eta \cdot Var_{x^{(t)}_i}(l_i^{(t)}-l_i^{(t-1)})$. I am confused over how does the $\frac{1}{2}$ factor arise. Note that this factor does matter in the proof of Theorem 3.1 (Eq. (104)-(106)). If the leading constant is indeed $1+O(1)$ instead of $\frac{1}{2}+O(1)$, the negative term in Eq. (104) would not be sufficient to cancel the second positive term. Perhaps the constant factor ($\frac{1}{2}$) in Lemma 4.2 should be made smaller?

2. I am also a bit confused over the proof of Lemma 4.2. In particular, how does on get from Eq. (96) to the base case of Eq. (97)? With $h=H-1$, the second term in Eq. (97) is $\mu^2\cdot \frac{2C_0\cdot H^2}{\alpha^3 \cdot \alpha_0^2H^6}=\mu^2\cdot polylog(T)$. Why is this greater than $\mu^2 T$?

Other comments:

3. The improvement in the dependence on $T$, compared to previous work on Hedge ($\tilde{O}(\sqrt{T})$) and Optimistic Hedge ($\tilde{O}(\sqrt{m}T^{1/4})$), seems to come at a cost of worse dependence on $m$, the number of players. In this sense, is the regret bound in this work really near-optimal?

4. Is there a trade-off between $m$ and $T$? If not, is it possible achieve $polylog(n,T)$ regret with no dependence on $m$?

5. In the current setting, player $i$ observes $E_{-i}\left[ \mathcal{L}_i(a_i, a^{-i})\right]$ for each $a_i$, instead of $\mathcal{L}_i(a_i,a^{-i})$. From a game-play point of view the latter feedback seems a bit more natural, as player $i$ don't have access to the randomness controlled by other players. I realize that existing literature also uses the former type of feedback, but is it the best formulation (from a game-play perspective)? Also, since the feedback is deterministic, isn't it possible to achieve regret independent of $T$?

**Time Spent Reviewing:**

5

---

> ### Author Response · Authors · 2021-08-10
> **Author Response**
>
> Thank you for your helpful suggestions.
>
> 1. There is no issue in the proof of Lemma 4.1, because the factor of 1/2 on the right-hand side of Eq. (30) ultimately leads to a factor of $\eta/2$ as opposed to $\eta$ in the term $\sum_t (\eta/2 + O(\eta^2)) \cdot Var_{x_i^{(t)}}(\ell_i^{(t)} - \ell_i^{(t-1)})$ of Eq. (27). In particular, the term $-\frac{1}{\eta} \sum_t KL(\tilde x_i^{(t)}; x_i^{(t)})$ is subtracted from $\sum_t \| x_i^{(t)} - \tilde x_i^{(t)} \|^\star_{x_i^{(t)}} \sqrt{Var_{x_i^{(t)}}(\ell_i^{(t)} - \ell_i^{(t-1)})}$ in Eq. (28) to lead to a factor of $\eta - \eta/2 = \eta/2$ in Eq. (27). This follows because the combination of Eqs. (30) and (33) show that $\frac{1}{\eta} \cdot KL(\tilde x_i^{(t)}; x_i^{(t)}) \geq (\eta/2 - O(\eta^2)) \cdot Var_{x_i^{(t)}}(\ell_i^{(t)} - \ell_i^{(t-1)})$ for each $t$.
>
> (As you mention, though, this factor of 1/2 is not really that important since the factor of 1/2 in Lemma 4.2 can be decreased to any positive constant with essentially no change in the proof.)
>
> 2. Thank you for pointing this out; there is indeed a typo in the proof of Lemma 4.2, with a factor of 2 missing from a few places. In particular the following changes to constants should be made:
>
> - In line 880, we need to set $\alpha_0$ to $\sqrt{\alpha/8}/H^3$ (as opposed to $\sqrt{\alpha/2}/H^3)$.
>
> - In Eq. (97), the term $(\alpha_0 \cdot H^3)^{2h}$ should instead be $(2 \alpha_0 \cdot  H^3)^{2h}$, so that in the base case there is an additional factor of $2^{2(H-1)} \geq T$ in Eq. (97).
>
> - Line 886 should say "The base case $h=H-1$ is verified by (96) and the fact that $2^{2(H-1)} \geq T$."
>
> - In the subsequent places of the proof of Lemma 4.2 where the term $(\alpha_0 \cdot H^3)^{2h}$ appears (specifically, lines 888 and 889, the derivation between lines 895-896, and line 896), again a factor of 2 should be added.
> (Observe that the inequality right before line 896 (and the equality in line 896) hold due to the updated setting of $\alpha_0 = \sqrt{\alpha/8}/H^3$.)
> The rest of the proof remains unchanged.
>
> 3. and 4. We do not know if the dependence on m in our regret bound for Optimistic Hedge can be improved (e.g., if polylog(n, T) regret is possible). We remark that we view the dependence on the other parameters (i.e., $n_i$ and $T$) as more interesting since the description length of a normal form game is exponential in the number of players m, so in most applications m should be small (e.g., a constant, as was considered by Chen & Peng, where m = 2). In games with special structure enabling description length polynomial in m (e.g., polymatrix games), we expect that better dependence on m could be proved using our approach.
>
> 5 For the setting you mention, namely where at each round each player i draws an action $a_i$ from their strategy vector $x_i^{(t)}$ for that round, and observes only the loss value $L_i(a_i, a_{-i})$, Rakhlin & Sridharan [1] conjectured that a regret better than sqrt(T) is not possible. We are not aware of any work on that problem, even for the simple case of 2-player zero-sum games. Whether the full-feedback model is well-suited for behavioral modeling depends on the application, but it is a standard model in the literature with many other applications, including its original use as an efficient computational method for solving for equilibria in games [2,3].
>
> In our setting (full feedback), we are not aware of any $\Omega(\log T)$ lower bound, so a regret independent of T for Optimistic Hedge is in principle possible.
>
> [1] A. Rakhlin & K. Sridharan. Optimization, Learning, and Games with Predictable Sequences, 2013.
>
> [2] G. Brown. Some Notes on Computation of Games Solutions. 1949.
>
> [3] J. Robinson. An Iterative Method of Solving a Game. 1951.

---

> > ### Comment · Reviewer_wnsJ · 2021-08-12
> > **Concerns addressed**
> >
> > Thank you for your detailed response. I no longer have concerns about the correctness of the result, and have increased my score accordingly.

---

### Official Review · Reviewer_dfrx · 2021-07-16

**Rating:** 10
**Confidence:** 5

**Summary:**

The paper studies no regret learning in multi-player general sum games. The major result of this paper is to prove that if every player adopts Optimistic Hedge, then individual regrets decays at rate $\tilde{O}(1/T)$, improve upon the previous $O(1/T^{5/6})$ of (Chen and Peng, NeurIPS 2020).

**Limitations And Societal Impact:**

Looks good.

**Main Review:**

Advantage: Learning in games is an important topics, and especially the convergence rate of no regret learning algorithms. The paper proves optimistic Hedge gives a near-optimal convergence rate of multi-player game, this is an important contribution. The technique of this paper is also very novel.

Drawback: To be picky, the writing of this paper could be improved, it would be better to explain the high level intuitions of the proof, e.g. why one wants to analyses the t-th order.

Summary: This is a very strong paper and I strongly recommend the acceptance of this paper. I didn't go through the proof, so I haven't checked the correctness of the proof. As I suggested above, it would be good to polish the writing, it would make reading easier.



**Time Spent Reviewing:**

1 hour

---

> ### Author Response · Authors · 2021-08-10
> **Author Response**
>
> Thank you for your helpful suggestions. We will add some additional explanation of the motivation behind analyzing higher-order differences.

---

### Official Review · Reviewer_2uVN · 2021-07-17

**Rating:** 9
**Confidence:** 4

**Summary:**

The paper shows that when all agents in a game use the Optimistic Hedge regret-minimization algorithm, their regret can be bounded as Otilde(1). When no assumption is made about what algorithm the other agents are using, the regret bound of Optimistic Hedge is still bounded as Otilde(sqrt(T)).

**Limitations And Societal Impact:**

I think it would be interesting to add a section describing, at a high level, what could be done to extend the positive result in the paper to more general settings. For example: perhaps some ideas from the paper can be extended to predictive OMD/FTRL on more general convex sets, or maybe it's possible to give a more general regret bound that works for any regularizer, akin to the RVU form of the regret bound given by Syrgkanis et al.

**Main Review:**

The paper is very clear and well-organized. It builds on several insights that were already known (for example, the analysis of FTRL in terms of local norms induced by the Hessian of the regularizer, in this case the negative entropy regularizer), and introduces new interesting tools to tighten the analysis of optimistic hedge. The paper cites all the relevant literature I was aware of.

The result is interesting and I think it would be a great addition to NeurIPS.

I have a couple of question for the authors:
1. Optimistic algorithms were historically developed as "predictive" algorithm, where at each time t the regret minimizer receives a prediction of the next loss before outputting its next decision/strategy $x^t$. It turns out that in most case it makes sense to set the prediction of the next loss to just be the last-received loss. The version of optimistic hedge considered in this paper can clearly be read as making that tacit assumption. However, the proofs in the paper by Syrgkanis et al. apply more generally to any prediction, giving a regret bound that scales with the norm of the mismatch between each loss and its prediction. I wonder if the same can be done in your paper. It would be interesting to figure out "how late" the assumption that prediction = latest loss can be made, and if a more general regret bound that scales with the norm of the mismatch can be given first, and only then under the assumption that prediction = latest loss the bound in the paper can be established. I find that keeping the prediction generic would help intuitively separate what is a general result, and what is "magic" that is made possible only by assuming something special about the prediction of the loss.

2. Is it possible to give a simple estimate for the magnitude of the constants involved in Theorem 3.1?

**Time Spent Reviewing:**

3

---

> ### Author Response · Authors · 2021-08-10
> **Author Response**
>
> Thank you for your helpful suggestions.
>
> 1. Generalizing to an arbitrary sequence of predictions is an interesting direction for future work. We expect that some analogue of Lemma 4.1 can be proven for an arbitrary sequence of predictions. Moreover, if the sequence of losses satisfies a sufficient "higher order smoothness property" (roughly speaking, that the higher order finite differences of the losses decay sufficiently quickly), then we expect that one could exploit the smoothness to establish stronger bounds on the regret.
>
> 2. A back-of-the-envelope calculation reveals that for the upper bound of the regret in Theorem 3.1, a constant of $10^8$ should suffice. We emphasize that we make no attempt to optimize constants and this could likely be substantially improved.
>
> 3. The question of extending our results to other regularizers and more general convex sets is an interesting direction for future work. We remark that for some regularizers, the dependence on some parameters will be worse than in our result (e.g., using self-concordant barriers would likely result in a $poly(n_i)$ dependence, whereas we get $\log(n_i)$).

---

### Decision · Program_Chairs · 2021-09-27

**Decision:**

Accept (Oral)

**Comment:**

This paper resolves a fundamental question on individual regret of
optimistic Hedge in a multi-player general-sum game setting, which was
left open since the work of Syrgkanis et al. 2015. Despite the recent
progress by Chen & Peng, 2020 which shows T^{1/6} regret, to achieve
polylog(T) regret this paper proposes significantly new ideas, which we
believe could be highly beneficial for the community. All reviewers are
excited about this result.